# DATA VALUATION AND SELECTION IN A FEDERATED MODEL MARKETPLACE

## ABSTRACT

In the era of Artificial Intelligence (AI), marketplaces have become essential platforms for facilitating the exchange of data products to foster data sharing. Model transactions provide economic solutions in data marketplaces that enhance data reusability and ensure the traceability of data ownership. To establish trustworthy data marketplaces, Federated Learning (FL) has emerged as a promising paradigm to enable collaborative learning across siloed datasets while safeguarding data privacy. However, effective data valuation and selection from heterogeneous sources in the FL setup remain key challenges. This paper introduces a comprehensive framework centered on a Wasserstein-based estimator tailored for FL. The estimator not only predicts model performance across unseen data combinations but also reveals the compatibility between data heterogeneity and FL aggregation algorithms. To ensure privacy, we propose a distributed method to approximate Wasserstein distance without requiring access to raw data. Furthermore, we demonstrate that model performance can be reliably extrapolated under the neural scaling law, enabling effective data selection without full-scale training. Extensive experiments across diverse scenarios, such as label skew, mislabeled, and unlabeled sources, show that our approach consistently identifies high-performing data combinations, paving the way for more reliable FL-based model marketplaces.

## 1 INTRODUCTION

With the rapid progress in AI, the acquisition of large-scale and high-quality datasets has become increasingly essential. In the past, traditional AI development frequently depended on easily obtainable web data, causing valuable yet isolated industry datasets to remain largely unused (Singh et al., 2024). To tackle this issue, data marketplaces have surfaced as vital platforms, facilitating broader data sharing and improving data accessibility, particularly in sectors with strict data regulations such as finance and healthcare. These marketplaces allow data buyers to access datasets from diverse sources, enhancing their research and application development. At the same time, data providers can monetize their data assets. Traditionally, the central platform plays a pivotal role in coordinating and facilitating these transactions. However, the acquisition of data within these marketplaces presents conspicuous challenges, spanning privacy risks, technical inefficiencies, and ethical deliberations.

Firstly, for buyers to make well-informed decisions when procuring data, they need to assess the quality and relevance of the datasets. However, this evaluation process is impeded by Arrow's Information Paradox (Arrow, 1972): data providers are hesitant to disclose data before payment due to the risk of unauthorized copying, while buyers require quality assessments prior to purchase (Lu et al., 2024). There is an urgent need for methods that enable data quality evaluation in a private way. Second, to ensure diversity, it is often preferable for buyers to acquire data from multiple providers, each offering data of varying quality and relevance. For instance, in the filed of automated driving, each provider may specialize in a particular type of vehicle, while the buyer needs a broad dataset covering various vehicle types for tasks. This requires efficient and effective methods for selecting and combining different data sources. Finally, current data marketplaces face significant challenges in preventing unauthorized redistribution and ensuring traceability of data ownership (Ranjbar Alvar et al., 2023). An emerging trend toward organization-wide sharing of data products, such as trained models, offers a way to address these challenges while enhancing data reusability (Mucci, 2024). Given challenges inherent in practical data marketplaces, our research is driven by following questions:

***(1) How can we determine the optimal selection strategy across multiple data sources when raw data cannot be shared?*** To achieve optimal performance for machine learning models, data buyers seek to acquire high-quality data from various potential sources and to determine the most effective combination strategy for these datasets. Traditional approaches necessitate either complete access to the data or partial observability of samples before a formal acquisition decision is made. However, in real-world data marketplaces, privacy concerns and cost constraints often preclude full data access, with a single sample frequently provided as a complimentary preview [1]. Recently, (Lu et al., 2024) proposes a linear experimental design approach to guide data acquisition. However, its practical applicability may be constrained by its dependence on the performance of a specific feature extractor and the restrictive assumption of linearity. (Li et al., 2024) proposes a model-agnostic method to evaluate data quality through distributional divergence in FL. While insightful for analyzing individual data sources, it doesn't directly quantify performance changes from specific data combinations. Similarly, (Chhachhi & Teng, 2024) differentiates data value based on individual data owners, neglecting correlations between datasets within clients, which limits its applicability in real-world scenarios.

***(2) How can we effectively leverage siloed data while preventing data misuse and enhancing data reusability?*** Data providers are now more acutely aware of the value of their data as well as the significant risks associated with uncontrolled data sharing (Zheng et al., 2022). Consequently, model trading has emerged as a potentially cost-effective and privacy-preserving paradigm for data marketplaces (Chen et al., 2019; Agarwal et al., 2019; Liu et al., 2021). Due to privacy concerns, data platforms often lack the capability to centralize siloed data from disparate sources for direct model training in response to service requests. Federated Learning(FL) (McMahan et al., 2017), a pivotal privacy-preserving technology, offers a compelling approach to circumvent this challenge. In the standard FL architecture, local clients (i.e., data sellers) collaboratively train a global model using their private data, thereby avoiding the direct disclosure of raw data. Instead, only model parameters or updates are exchanged with a central platform (i.e., the platform), which aggregates these contributions iteratively to refine the global model until convergence is achieved.

In our setting, the platform collaborates with data sellers to train models without direct access to their raw data, thereby preserving data privacy. This shift from a centralized to a federated approach introduces new technical challenges due to the heterogeneity of data distributions across sources:

*Technical Challenge 1* In the FL setting, enumerating all possible data combinations to determine the optimal acquisition strategy becomes infeasible—especially when there are many sellers, each holding substantial amounts of data. While the Wasserstein distance has been used as a proxy for centralized model performance (Just et al., 2023; Kang et al., 2024), we find that such an estimator fails under the non-i.i.d. conditions common in FL. This calls for a new Wasserstein-based performance estimator, as well as an efficient method for computing it in a distributed fashion.

*Technical Challenge 2* Even with a well-selected data combination, model performance still depends critically on the choice of aggregation algorithm. For instance, a buyer seeking a model capable of recognizing a wide range of labels may need to acquire data from sources that each contain only a subset of labels. While this strategy improves label coverage, the resulting data heterogeneity introduces substantial challenges to FL model convergence. In such cases, local models may diverge significantly due to differing data distributions, causing simple aggregation methods to produce suboptimal global models. Thus, even when the acquired data aligns well with the target task, a mismatch between data heterogeneity and the chosen FL algorithm can undermine overall performance. This underscores the importance of selecting aggregation strategies that are robust to heterogeneity *before* formal training.

The main contributions of this paper are as follows: (1) This study provides a practical solution for FL model marketplaces, which simultaneously addresses performance prediction, projection, and optimal data mixture without the need for costly and time-consuming full-scale training runs; (2) We propose a novel Wasserstein-based performance estimator, CombineWad, tailored to FL settings. CombineWad provides reliable performance prediction across diverse data combinations without requiring full-scale training; (3) We further demonstrate that CombineWad not only predicts model performance but also serves as a strong signal for evaluating the compatibility between data heterogeneity and FL aggregation algorithms. This allows buyers to assess whether a chosen algorithm is robust to the

---

[1]https://datarade.ai/data-categories/ai-ml-training-data

acquired data distribution prior to initiating full-scale training; (4) To ensure privacy in federated settings, we develop an efficient and privacy-preserving method for approximating Wasserstein distance in a distributed manner; (5) We conduct extensive experiments across various applications to demonstrate the effectiveness of the proposed framework, paving the way for building more reliable and trustworthy model marketplaces.

## 2 TECHNICAL PRELIMINARIES

We provide preliminaries of the Wasserstein Distance for better understanding main techniques. Due to space limit, please refer to Appendix A for related work of *Federated Learning*, *Integration of Federated Learning and Optimal Transport*, *Data valuation and acquisition*.

**Definition 2.1.** (Wasserstein distance) The $p$-Wasserstein distance between measures $\mu$ and $\nu$ is

$$\mathcal{W}_p(\mu, \nu) = \left( \inf_{\pi \in \Pi(\mu,\nu)} \int_{\mathcal{X} \times \mathcal{X}} d^p(x, x') d\pi(x, x') \right)^{1/p}, \tag{1}$$

where $d^p(x, x')$ is the pairwise distance metric such as $d^p(x, x') = \|x - x'\|_p$. $\pi \in \Pi(\mu, \nu)$ is the joint distribution of $\mu$ and $\nu$, and any transportation plan $\pi$ attains such minimum is considered as an *Optimal Transport* plan. In the following paper, we focus on $p = 2$, and omit the $p$ for simplicity.

In the discrete space, the two marginal measures are denoted as $\mu = \sum_{i=1}^m a_i \delta_{x_i}, \nu = \sum_{j=1}^n b_j \delta_{x'_j}$ ,where $\delta_{x_i}$ is the dirac function at location $x_i \in \mathbb{R}^d$, and $a_i, b_j$ are probability masses such that $\sum_{i=1}^m a_i = \sum_{j=1}^n b_j = 1$. Therefore, the Monge problem (Monge, 1781) seeks a map that must push the mass of $\mu$ toward the mass of $\nu$. However, when $m \neq n$, the Monge maps may not exist between a discrete measure to another, especially when the target measure has larger support size of the source measure (Peyré et al., 2019). Therefore, we consider the Kantorovich's relaxed formulation (Kantorovitch, 1958), which allows *mass splitting* from a source to several targets as

$$\mathcal{W}(\mu, \nu) = \min_{\mathbf{P} \in \Pi(\mu, \nu)} \langle \mathbf{C}, \mathbf{P} \rangle \tag{2}$$

where $\mathbf{C} = \left[ \|x_i - x'_j\|_2 \right]_{i,j=1}^{m,n}$ is the pairwise Euclidean distance matrix, and $\Pi(\mu, \nu) = \{\mathbf{P} \in \mathbb{R}_+^{m \times n} | \mathbf{P}\mathbf{1}_m = \mu, \mathbf{P}^\top \mathbf{1}_n = \nu\}$ is the set of all transportation couplings.

Such OT problem is a constrained convex minimization, which is naturally paired with a dual problem (constrained concave maximization problem) as follows

$$\mathcal{W}(\mu, \nu) = \max_{(f,g) \in \mathcal{R}(d)} \langle f, \mu \rangle + \langle g, \nu \rangle, \tag{3}$$

where $\mathcal{R}(d) = \{(f, g) \in \mathcal{C}(\mathcal{X}) \times \mathcal{C}(\mathcal{X}) : \forall (x, x'), f(x) + g(x') \leq d(x, x')\}$, $\mathcal{C}$ is a collection of continuous functions.

With the abuse of notations, suppose $\mathcal{W}(f^\star, g^\star)$ be the objective value with the optimal dual solutions $f^\star$ and $g^\star$. Based on (Just et al., 2023), the gradient of the Wasserstein distance w.r.t. the probability mass of data points in the two datasets can be expressed as

$$\nabla_\mu \mathcal{W}(f^\star, g^\star) = (f^\star)^T, \ \nabla_\nu \mathcal{W}(f^\star, g^\star) = (g^\star)^T. \tag{4}$$

Furthermore, the *calibrated gradient* introduced by (Just et al., 2023) could predict how the Wasserstein distance changes as more probability mass is shifted to a given data point $z_i$ in $\mu$, as follows

$$\frac{\partial \mathcal{W}(\mu, \nu)}{\partial \mu(z_i)} = f_i^\star - \sum_{j \in \{1, \cdots, m\} \backslash i} \frac{f_j^\star}{m - 1}. \tag{5}$$

Therefore, when $\mu$ refers to a training set to be evaluated, $\nu$ refers to a clean validation set, this gradient is a power tool to detect and prune abnormal or irrelevant data points in the training set. Specifically, data points with higher gradient score are considered noisy.

## 3 MARKET DESCRIPTION

To support a clearer understanding of the technical content, we provide Table 1, which summarizes the important notations used throughout this work.

**Model Buyer and Selection Decision**: Suppose a model buyer holds a validation dataset $D^{\text{val}}$, which represents the target data distribution. To achieve satisfactory performance on this target distribution, the buyer seeks to obtain a model $\mathcal{M}$ trained with data from multiple sources under the FL setting. The performance of this model is measured by a metric $V$, which takes a trained model and a validation dataset to produce a performance score. Thus, the performance of a model trained on other datasets and evaluated on $D^{\text{val}}$ is expressed as $V(\mathcal{M}(\cdot), D^{\text{val}})$. For the remainder of this paper, we will simplify this notation to $V(\cdot, D^{\text{val}})$. Given a budget of $N$ samples, the buyer must determine a *mixing ratio* $\mathbf{p} = \{p_1, \ldots, p_m\}$, where each $p_i$ denotes the proportion of the budget allocated to data provider $i$, subjecting to the constraint $\sum_{i=1}^{m} p_i = 1$. The resulting training dataset is denoted as $D(N, \mathbf{p}) = \bigcup_{i=1}^{m} D_i^{\text{tr}}$, where each subset $D_i^{\text{tr}}$ is a randomly selected portion of the seller's full dataset $D_i^{\text{all}}$, and the size of each subset is constrained by $|D_i^{\text{tr}}| = p_i N$. The buyer has two primary goals for acquisition (Kang et al., 2024):

(1) **Performance Maximization under a Fixed Budget**: Given a constrained acquisition budget $N$, the buyer aims to maximize model performance by optimally selecting the mixing ratio $\mathbf{p}$. This objective can be formulated as $\max_{\mathbf{p}} V(D(N, \mathbf{p}), D^{\text{val}})$.

(2) **Cost Minimization for a Target Performance**: The buyer seeks to minimize the data selection budget $N$ required to achieve a target performance level $\tau$, by jointly choosing $N$ and $\mathbf{p}$. The objective is expressed as $\min_{N, \mathbf{p}} N$ with the constraint $V(D(N, \mathbf{p}), D^{\text{val}}) \geq \tau$.

**Data Provider**: Suppose there are $m$ prospective data providers, each holding a dataset denoted by $D_1^{\text{all}}, \cdots, D_m^{\text{all}}$. We consider an FL setting in which only the model trained on multiple sources is made available for transactions. Therefore, all raw data remain private and cannot be shared. Directly optimizing $\mathbf{p}$ requires training FL models with different combinations of data. This is challenging and computationally expensive when the size of $N$ is large. To address this issue, only small samples from each data source are made available for the model buyer to make selection decisions. We refer to these samples as *pilot data*, denoted by $D_i^{\text{pi}}$, where $|D_i^{\text{pi}}| \ll |D_i^{\text{all}}|$. Each provider $i$ will take part in the *federated trial runs* using their pilot data. After completing these trial runs, each provider $i$, upon accepting the *training request* for acquiring $p_i N$ samples, will randomly sample a subset $D_i^{\text{tr}}$ from $D_i^{\text{all}}$ for the formal federated training. We assume these sampling subsets follow the same distribution as the whole dataset.

To illustrate the potential of federated trial runs, we present a toy example in which the FedProx algorithm (Li et al., 2020) is applied to the CIFAR-10 dataset under a label-skewed setting. Model accuracy is evaluated on a balanced validation set across varying training budget sizes. As shown in Figure 1, each line corresponds to a different data mixing ratio. Notably, mixing ratios that yield high accuracy (green) with smaller training budgets tend to maintain strong performance as the training set size increases. This result indicates that preliminary trials can effectively identify the optimal $\mathbf{p}^\star$, allowing a reliable performance projection in larger data sets using the same configuration.

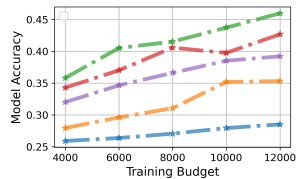

Figure 1: High-accuracy mixing ratios scale well

**Central Platform**: Suppose there exists a trusted central platform that orchestrates the model aggregations in federated training with data sellers, provides the black-box API for the model buyer during trial runs, and helps to make the data selection decision.

## 4 METHODOLOGY

Our methodology incorporates an initial phase focused on evaluating potential model performance and optimizing data utilization strategies before full-scale training. Figure 2 illustrates the overall workflow, comprising two core components: (1) evaluating model performance through trial runs on pilot data, and (2) precisely predicting model performance and optimizing data mixing ratios. set. To address the computational challenge of exploring numerous mixing ratios with limited pilot data, we introduce a novel approach leveraging the Wasserstein distance. Specifically, Sec 4.1 proposes using the Wasserstein distance as a surrogate to predict model performance across various data compositions and scales, thus reducing the need for extensive training. Sec 4.2 presents an efficient method for privately computing this distance. Consequently, trial runs involve calculating the

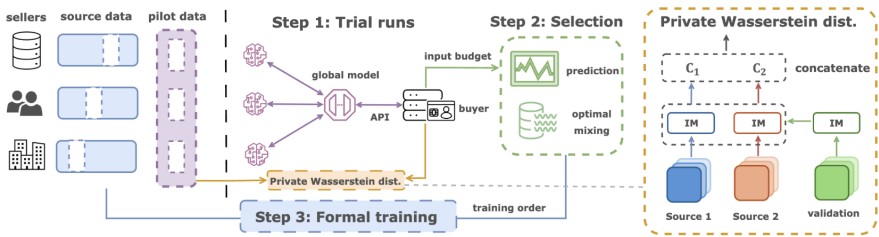

Figure 2: Overall Framework. (1) During trial runs, each client performs local training on a subset of their private pilot data based on a given mixing ratio. The platform then aggregates these models and provides an API for model buyers to assess performance on their validation. (2) Then the platform construct the performance estimator and determine the optimal mixing ratio. (3) After determining the mixing ratio, the platform orchestrates a formal FL training.

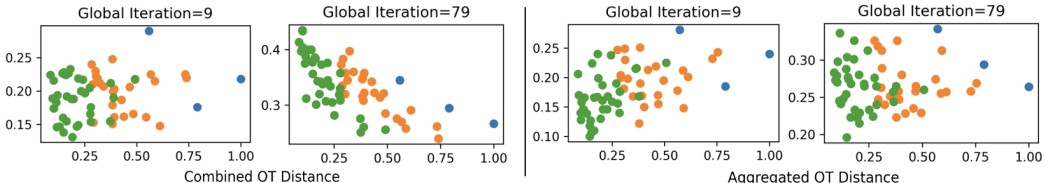

Figure 3: CombineWad offers a better proxy of model performance than AggWad in the FL setting; Green, orange and blue dots represent models trained with three sources, two sources and single source. **Left**: Accuracy vs CombineWad; **Right**: Accuracy vs AggWad.

Wasserstein distance between pilot data combinations and the validation data. Subsequently, Sec 4.3 explains how the platform builds a performance estimator using this distance as input to predict model performance under different data scales and compositions, enabling fine-grained optimization of the data selection strategy. The algorithmic details are provided in the Appendix.

## 4.1 SMALLER COMBINED WASSERSTEIN DISTANCE ENABLES BETTER VALIDATION PERFORMANCE

Wasserstein distance has been proved as an upper bound for measuring the discrepancy between the training and validation performance of in the field of the domain adaptation (Just et al., 2023; Kang et al., 2024; Courty et al., 2016; Redko et al., 2017; Montesuma & Mboula, 2021; Chhachhi & Teng, 2024; Li et al., 2024). However, in FL the global model is the combination of multiple local models, building this upper bound relationship with multi-source distributions, especially when heterogeneous data distribution, has significant affects on the FL model convergence.

We investigate non-i.i.d. data distributions across three clients and analyze the relationship between the class-wise Wasserstein distance and the model's validation performance, as this distribution property is more challenging and common in real-world scenarios. In this setup, each client possesses partial and non-overlapping labels (an i.i.d. setting is detailed in Appendix C.2). Therefore, a proper combination of these sources is desirable to ensure the label diversity. We explore two different calculations. (1) *AggWad*: which represents the weighted aggregation of pairwise Wasserstein distances, defined as $\sum_{i=1}^{m} \alpha_i \mathcal{W}(D_i^{\mathrm{pi}}, D^{\mathrm{val}})$. This approach assesses each data source's quality and quantity in isolation, neglecting any connections between them. (2) *CombineWad*: which calculates the Wasserstein distance using the combined data from all sources, denoted by $\mathcal{W}(\sum_{i=1}^{m} \alpha_i D_i^{\mathrm{pi}}, D^{\mathrm{val}})$. As illustrated in Figure 3, our empirical findings reveal an initial negative correlation between validation performance and AggWad at the beginning of training (epoch 9), suggesting the global model's tendency to be influenced by local models. In contrast, as the model converges (epoch 79) and incorporates information across all clients, CombineWad exhibits a strong correlation with the validation performance. These empirical observations are theoretically grounded in Theorem 4.1.

**Theorem 4.1.** *We denote $f_t^i, f_v$ as the labeling function for training and validation data. Let $f_t^i(\cdot)$ be the $i$-th local model and $f(\cdot)$ be the aggregated global model. Let $\{\mu_t^i\}_{i=1}^{m}, \mu_v$ be the training and validation distribution. Suppose that the loss function $\mathcal{L}$ is $k$-Lipschitz, and define*

$\mathcal{L}^{val}(\theta) = \mathbb{E}_{\mu_v(x)}\big[\mathcal{L}\big(f_v(x), f(\theta, x)\big)\big]$, *then we have*

$$\mathcal{L}^{val}(\theta) - \mathcal{L}^{ERM}(\theta) \leq \mathcal{W}\big(\sum_{i=1}^{m} \alpha_i \mu_t^i, \mu_v\big) + k\big(\mathcal{L}^{ERM}(\theta^\star) + \mathcal{L}^{val}(\theta^\star)\big),$$

*where* $\quad \mathcal{L}^{ERM}(\theta) = \sum_{i=1}^{m} \alpha_i \mathbb{E}_{\mu_t^i(x)} \mathcal{L}\big(f_t^i(x), f(\theta, x)\big), \theta^\star = \arg\min \big\{ \mathcal{L}^{ERM}(\theta) + \mathcal{L}^{val}(\theta) \big\}.$

The proof is shown in Appendix B.1. There are several advantages in taking CombineWad as the surrogate of the validation performance: Firstly, the theorem demonstrates that the model's validation performance is bounded by an affine transformation of the CombineWad. Consequently, once this transformation has been learned, we can directly predict the model performance via such transformation without enumerating all potential mixing ratios to train a multitude of federated models. Moreover, the linear characteristics of the optimal transport problem allow for a sensitivity analysis, facilitating a more fine-grained optimization of the mixing ratio. We will dive deeper into the above two advantages in Section 4.3. More interestingly, we find CombineWad could serve as a convergence signal to check whether a specific FL aggregation algorithm could handle the data heterogeneity well. We conduct extensive experiments in Section 5.1 to validate this hypothesis.

## 4.2 PRIVATE-ENHANCED FEDERATED WASSERSTEIN DISTANCE

Calculating the Wasserstein distance typically requires raw data access, which is infeasible in our privacy-preserving setting. While Differential Privacy (DP) is a standard approach to ensuring privacy, the error introduced by DP can be relatively significant, as reported in (Lê Tien et al., 2019). Recent Federated Wasserstein distance approximations (Rakotomamonjy et al., 2024; Li et al., 2024) rely on sharing interpolating measures and iterative triangle inequality applications, suffering from high costs and single-seller limitations. Our work tackles the more relevant multi-seller scenario, requiring aggregation before computing the distance to validation data (Section 4.1).

Our approach leverages these geometric properties for a more efficient and suitable Wasserstein distance estimation in our multi-seller context. To illustrate the technique, we begin with the case of a single seller versus a single buyer. Subsequently, we will elaborate how to extend this approach to multiple sources. Consider two data sets $D^{pi}$ with data size $n$ and $D^{val}$, which are held by the data seller and the model buyer respectively, and a randomly initiated and global shared measure, e.g. gaussian, $\mathbf{x}^\gamma \in \mathbb{R}^{k \times d} \sim \mathcal{N}(m_\gamma, \sigma_\gamma^2)$. By applying the barycentric mapping (Courty et al., 2018), an interpolating measure $\eta^{pi}(t)$ is the interpolation between raw data $D_i^{pi}$ and the global $\gamma$ via

$$\eta^{pi}(t) = \frac{1}{n} \sum_{i=1}^{n} \delta_{(1-t)x_i^{pi} + tn(\mathbf{P}^\star(D^{pi}, \gamma)\mathbf{x}^\gamma)_i}, \; x_i^{pi} \sim D^{pi}, \tag{6}$$

where the $t \in [0, 1]$ is the push-forward parameter that controls "how much" the source data $D^{pi}$ is pushed forward to the target data $\mathbf{x}^\gamma$. $\mathbf{P}^\star(D^{pi}, \gamma) \in \mathbb{R}^{n \times k}$ is the OT plan between $D^{pi}$ and $\mathbf{x}^\gamma$. Constructing $\eta^{val}(t)$ follows a similar procedure. Then we could approximate $\mathcal{W}(D^{pi}, D^{val})$ via

$$\hat{\mathcal{W}}(D^{pi}, D^{val}) = \frac{1}{1-t}\mathcal{W}(\eta^{pi}(t), \eta^{val}(t)). \tag{7}$$

**Theorem 4.2.** *The approximation error* $|\hat{\mathcal{W}}(\mathbf{x}^\mu, \mathbf{x}^\nu) - \mathcal{W}(\mathbf{x}^\mu, \mathbf{x}^\nu)|$ *is bounded by* $\mathcal{O}(c\sigma_\gamma)$, *where c is a small constant associated with k, which is the data size of the global share measure* $\mathbf{x}^\gamma$. *Specifically, this approximation error will decrease with rate* $\mathcal{O}(\frac{1}{k})$ *when k increases.*

Theoretical proof and empirical validation are shown in Appendix B.2. We further modify it to enable the proposed technique to calculate distances among multiple data sources. The key idea is to directly combine all of pairwise distance matrices to construct a larger one, which can then be employed as the input for the OT problem. Following the similar procedure, $\mathbf{x}^\gamma$ is randomly initialized and shared with all data sellers and the buyer. The buyer constructs $\eta^{val}(t)$ and sends it to the platform. Simultaneously, the $i$-th seller constructs own $\eta_i^{pi}(t)$ and sends it to the platform. After collecting all interpolating measures, the platform calculates the point-wise euclidean distance matrix for each pair of $\{\eta_i^{pi}(t), \eta^{val}(t)\}$ as $\mathbf{C}_{pi}^i = \mathbf{C}\big(\eta_i^{pi}(t), \eta^{val}(t)\big) = \big[\|x_j - x_l'\|_2\big]_{j,l=1}^{n_i, k}$, where $x_j \sim$

$\eta_i^{\text{pi}}(t), x'_l \sim \eta^{\text{val}}(t), n_i = |D_i^{\text{pi}}|$. The new cost matrix is constructed via $\mathbf{C}_{\text{pi}} = \left[ \mathbf{C}_{\text{pi}}^i, \cdots, \mathbf{C}_{\text{pi}}^m \right]$, where $\mathbf{C}_{\text{pi}} \in \mathbb{R}^{[\sum_{i=1}^m n_i] \times k}$ is utilized as an input to optimize the OT problem, and we could approximate approximate $\hat{\mathcal{W}}(\sum_{i=1}^m D_i^{\text{pi}}, D^{\text{val}}) = \frac{1}{1-t} \min_{\mathbf{P}} \langle \mathbf{C}_{\text{pi}}, \mathbf{P} \rangle$.

## 4.3 PERFORMANCE PREDICTION, PROJECTION, AND DATA SELECTION

In the previous Sec 4.1, we have verified that the federated model's validation performance is bounded by an affine transformation of the CombineWad. Therefore, we could also conduct multiple trial runs (very low budget) to learn such a transformation and then predict the model performance with any mixing ratio and any data size.

Each performance estimator provides a light way to approximate the model performance with any mixing ratio on a specified data scale $N_j$. In order to predict the performance when the data scale attains at the specified acquisition budget $N$ in the formal training, we leverage the theoretical analysis from (Kang et al., 2024), which leverages the neural scaling laws, and projects the performance for a particular distribution onto larger data scales. Assume one has completed the fitting of the performance predictor $\hat{V}_i(D(N_i, \mathbf{p}), D^{\text{val}}), \hat{V}_j(D(N_j, \mathbf{p}), D^{\text{val}})$ on two different scales $N_i < N_j$, then the model performance for any data mixture $\mathbf{p}$ at any data scale $N$ can be predicted as

$$\hat{V}(D(N, \mathbf{p}), D^{\text{val}}) = \left( \log \frac{N_j}{N_i} \right)^{-1} \left[ \log \frac{N}{N_i} \hat{V}_j - \log \frac{N}{N_j} \hat{V}_i \right]. \tag{8}$$

Therefore, by performing the fitting process at different small scales for once, we do not need to fit any additional parameters for a large data scale. This is particularly beneficial for reducing computational overheads, as well as meeting the acquisition goals of the model buyer, who wants to obtain a model with a desired performance with minimum acquisition costs.

Until now, we have discussed techniques of predicting and projection the model performance, and how to leverage the Wasserstein distance as the signal to guide the data selection. However, in order to find the optimal mixing ratio $\mathbf{p}^\star$, it is necessary to explore how perturbations in the mixing ratio can impact the model's performance. We start with a randomly initialization with $\mathbf{p} = \mathbf{p}^0$ such that $\mathbf{p}$ remains within the simplex $\sum_{i=1}^m p_i = 1$. Then we carry out the iterative procedure similarly as (Kang et al., 2024)

$$\mathbf{p}^{(t+1)} \leftarrow \mathbf{p}^{(t)} + h_t \left. \frac{\partial \hat{V}(D(N, \mathbf{p}), D^{\text{val}})}{\partial \mathbf{p}} \right|_{\mathbf{p}=\mathbf{p}^{(t)}}, \tag{9}$$

where $h_t$ is the step size at iteration $t$. As the performance estimator incorporates the Wasserstein distance as a proxy, it further requires the gradient score w.r.t. the Wasserstein distance $\frac{\partial \mathcal{W}(D(N, \mathbf{p}), D^{\text{val}})}{\partial \mathbf{p}}$. Thanks to the development of the calibrated gradient as in equation 5, we could predict how the Wassserstein distance changes if upweighting a training dataset (more probability mass is shifted to that dataset). Such gradients are easily available as during the calculation of the Wasserstein distance, where we could simultaneously obtain its dual solutions as shown in equation 3 and equation 4.

## 5 EXPERIMENTS

In this section, our evaluations are threefolds: *(1)Model Convergence Assessment*, where CombineWad could serve as a predictive signal to assess the model performance and model convergence. *(2) Performance Prediction*, where for any mixing ratio of data sources and any data scale, we could predict the performance of the model trained on a given composed dataset. *(3) Optimal Data Selection*, where for a given data budget, we find a mixing ratio of data sources that can maximize the performance of a model. For all experiments, we set up the problem with three data sources, where each source consists of different classes, to simulate the non-i.i.d setting. *(4) Private Wasserstein Distance*, where for multiple datasets distributed in multiple parties, our method could provide relatively accurate approximations without sharing raw data.

**FL aggregation algorithms**. We implement four representative FL algorithms to train models: FedAvg (McMahan et al., 2017), FedProx (Li et al., 2020), Scaffold (Karimireddy et al., 2020), and FedNova (Wang et al., 2020).

**Datasets**. We use CIFAR10, MNIST, Fashion, ImageNet and one real-world medical dataset RSNA Pediatric Bone Age (Halabi et al., 2019) for evaluations.

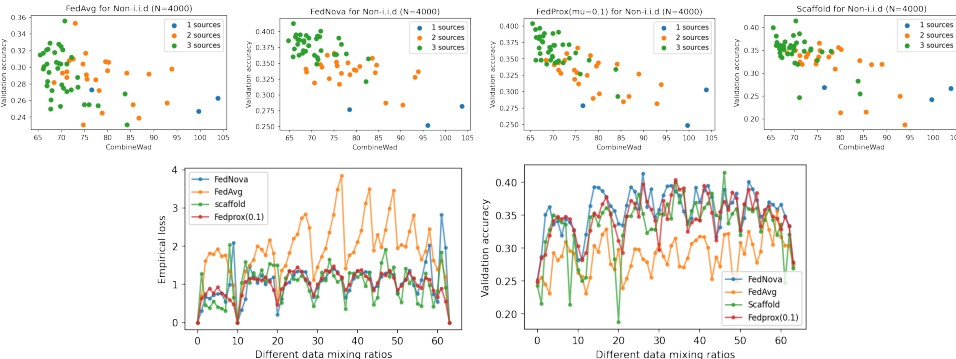

Figure 4: A comparison of FedAvg, FedNova, FedProx and Scaffold in a three-source setting ($N = 4000$). FL models with lower training loss and better validation performance has a more distinct correlation between validation performance and CombineWad.

### 5.1 COMBINEWAD AS A PREDICTIVE SIGNAL TO ASSESS THE MODEL PERFORMANCE

To validate the efficacy of CombineWad as an indicator of the suitability of FL aggregation algorithms, we conduct a comparative study involving FedAvg, FedProx, SCAFFOLD, and FedNova under a skewed label distribution. The data simulation is designed with three distinct data sources, each containing a partial and non-overlapping subset of the full label space. For a buyer aiming to obtain a balanced training dataset with comprehensive label coverage, acquiring data from all three sources represents the most desirable strategy.

Our analysis of the training dynamics across these algorithms revealed notable differences in the correlation between CombineWad and model accuracy. The top panel of Figure 13 presents four scatter plots, each illustrating the relationship between validation accuracy and varying scales of CombineWad for one of the four FL algorithms. Each plot contains 64 points, corresponding to different data mixing ratios. The bottom panel of Figure 13 displays the empirical training loss and validation accuracy of the four FL algorithms, where the x-axis denotes the index of each mixing ratio. Notably, FedAvg exhibits a weak negative correlation between CombineWad and validation accuracy: training with data from all sources (i.e., lower distances) does not consistently yield better performance compared to training with only two or even a single source. In contrast, FedProx demonstrates a relatively strong and consistent negative correlation, where lower CombineWad values are consistently associated with higher validation accuracy. Specifically, FedProx achieves lower training loss, reduced loss variance, and higher validation accuracy than the other algorithms. These findings support the hypothesis that CombineWad serves as a predictive signal for assessing model performance. Additional experiments and discussions are provided in Appendix E.

### 5.2 PERFORMANCE PREDICTION AND DATA SELECTION

For the task of performance prediction, we fit the parameters on limited compositions and extrapolate the predictions to unseen compositions. This is to demonstrate the effectiveness of leveraging the combined Wasserstein distance. For the task of performance projection, we aim to predict the model performance at an unseen larger data scale, based on the model performance of a small data size. More details of baselines are shown in Appendix D.2.

**Performance Prediction** We conduct the federated training for 3 data sources with a pre-specified training budget. We randomly partition the data into training and testing subsets at a ratio $70\%$ for training and $30\%$ for testing. We measure the correlation of the predicted and actual performance in Fig 5. Compared to other baselines, our estimator is more accurate with $r^2 = 0.97$ for the training data and $r^2 = 0.74$ for the testing data. This outcome underscores the robust representational capacity of leveraging the combined Wasserstein distance. The reason for the poor performance of other baselines comes from the non-i.i.d setting in the federated training.

**Performance Projection** To verify the effectiveness of the neural scaling law equation 8 in the context of FL, we conduct trial runs on CIFAR10 and MNIST with two training budgets $N_0 = 4K, N_1 = 8K$ respectively, and extrapolate the performance to a larger undisclosed data size $N = 15K$. We

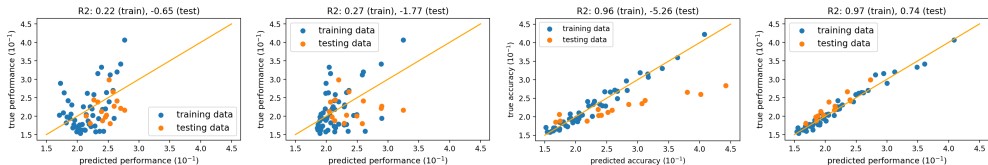

Figure 5: Predicted model performance vs. true model performance for extrapolation for 3 data sources. From upper left to bottom right are: Linear, Pseudo-Quadratic, CombineWad, Ours.

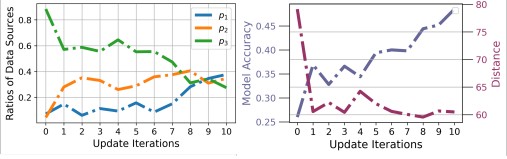
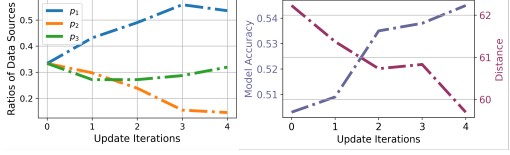

(a) When there are highly label skews among sellers, the mixing ratio converges to the uniform distribution (matching the balanced validation set).

(b) The mixing ratio assigns lower weights to the data sources containing mislabeled data (mislabeled proportions: 20% in source 2 has and 5% in source 3).

Figure 6: Data selections for ImageNet (left) and CIFAR10 (right) under different scenarios.

compare the true accuracy and the predicted accuracy in Appendix 9. For both experiments, we can achieve high correlation scores ($\approx 0.88$), showing the promise of the neural scaling law in FL.

In this section, we will explore whether the proposed selection strategy could help to find the best mixing ratio. Specially, we are facing the problem of choosing $N = 15K$ samples in total to train the formal model, with the pilot dataset of $N_i = 5K$ and $N_j = 8K$ for trial runs. Our evaluation consists of two phases: In the first phase, we predict the model performance of 15K samples via equation 8. Then we iteratively update $\mathbf{p}^t$ via equation 9. It is worthy to note that during this procedure, we do not train any FL model for the formal training. In the second phase, we evaluate each $\mathbf{p}^{(t)}$ by actually training the model and evaluate the validation performance and the distance.

**Data Selection with highly label skews among sellers** The first case is that the validation set is a balanced set, while each source has only partial and non-overlapping labels, showing highly label skews. Suppose the mixing ratio is initialized randomly as $\mathbf{p}^0 = \{0.08, 0.06, 0.86\}$. In each iteration, we record the updated mixing ratio of data sources, the corresponding model accuracy and Wasserstein distance. As shown in Figure 6a, the mixing ratio converges to almost the uniform distribution. As a result, the selected training set will have a comparable number of samples for each label, aligning its distribution with that of the balanced validation set. In the right panel, the model performance continuously increases and the constructed training data has smaller Wasserstein distance with the validation data during the update iterations.

**Data Selection with Mislabeled data** The second case is that some data sources contain mislabeled data. To simulate such a setting, we first establish an i.i.d. data distribution across all sources, ensuring each initially has the same label distribution. Then we randomly mislabel 20% data in source 2, and 5% data in source 3. Suppose the mixing ratio is $\mathbf{p}^0 = \{1/3, 1/3, 1/3\}$. As shown in Figure 6b, the mixing ratio assigns higher weight to source 1, which is clean data, and assigns lower weights to source 2 and source 3. The higher the proportion of mislabeled data, the lower the weight. This adaptive re-weighting mechanism effectively reduces the influence of mislabeled data in the selected training set, leading to a continuous increase in model accuracy.

## 6 CONCLUSION

In this work, we present a general framework for data selection and performance prediction in federated model marketplaces to promote data sharing. Our approach enables optimal data selection from multiple sources without revealing raw data and estimates federated model performance using the Wasserstein distance and neural scaling laws—without requiring actual training. This method offers a foundation for trustworthy model delivery and data value quantification. However, due to approximation errors in computing the Wasserstein distance, the optimal mixing ratio may fluctuate during evaluation. A more robust estimation method is needed to address this issue. Furthermore, our framework can be extended to incorporate a pricing mechanism, which we leave as future work.

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

| Notations | Descriptions |
|---|---|
| $D_i^{\text{all}}$ | All data held by $i$-th data provider |
| $S_i$ | The subset of pilot data from $i$-th data provider for one trial run |
| $D_i^{\text{pi}}$ | The whole set of pilot data provided by $i$-th data provider |
| $D^{\text{val}}$ | Validation data provided by the model buyer |
| $D_i^{\text{tr}}$ | Training data provided by $i$-th data provider for formal training |
| $N$ | The training budget |
| $\mathbf{p} = \{p_i\}_{i=1}^m$ | The data mixing ratio |
| $D^{\text{tr}}(N, \mathbf{p}) = \cup_{i=1}^m D_i^{\text{tr}}$ | Training data for the formal training |
| $D^{\text{pi}}(N, \mathbf{p}) = \cup_{i=1}^m S_i$ | Training data for one trial run |
| $V(\cdot, D^{\text{val}})$ | The validation performance on $D^{\text{val}}$ when trained on the candidate data |
| $\mathcal{W}(\cdot, D^{\text{val}})$ | The Wasserstein distance metric |
| $B_s$ | The budget of the trial runs |
| $\gamma$ | Randomly initialized and global shared measure |
| $\eta_i^{\text{pi}}$ | Interpolating measure between $D_i^{\text{pi}}$ and $\gamma$ |
| $\eta^{\text{val}}$ | Interpolating measure between $D^{\text{val}}$ and $\gamma$ |
| $\mathcal{I}_i^j$ | The index of the sampled $S_i$ in $j$-th trial run |
| $\mathbf{C}_{\text{pi}}$ | The concatanated cost matrix, $\mathbf{C}_i = \|\eta_i^{\text{pi}} - \eta^{\text{val}}\|$ |

Table 1: Table of Notations

# A  MORE RELATED WORK

## A.1  FEDERATED LEARNING

Federated Learning (FL) is a distributed learning framework that enables massive and remote clients to collaboratively train a high-quality central model. This paper focuses on cross-silo Federated Learning scenarios involving up to hundreds of clients, wherein each client possesses a substantial volume of data. The objective function of FL takes the form of an Empirical Risk Minimization (ERM) as

$$\mathcal{L}^{ERM}(\theta) = \sum_{i=1}^m \alpha_i \mathcal{L}_i^{\text{ERM}}(\theta),$$
$$\mathcal{L}_i^{\text{ERM}}(\theta) = \mathbb{E}_{x \sim D_i}[\ell_i(\theta, x)], \ \sum_{i=1}^m \alpha_i = 1. \tag{10}$$

where $\theta \in \mathbb{R}^d$ represents the parameter for the global model, $\ell_i(\theta, x)$ are the local loss functions, which are often identical between all clients.

FedAvg (McMahan et al., 2017) has been a de facto algorithm for FL, which aggregated the model by simple averaging. However, the distribution of each local dataset is highly different from the global distribution, thus the local objective of each party is inconsistent with the global optima. This non-i.i.d property can exert a significant impact on the accuracy of FedAvg. There have been several research efforts aimed at tackling the statistical heterogeneity in FL, with the intention of ensuring that the averaged model remains in closer proximity to the global optimum (Wang et al., 2020; Li et al., 2020; Karimireddy et al., 2020; Acar et al.). Our work does not aim to develop algorithms to address heterogeneous or adversarial distributions in FL (Mohri et al., 2019; Cho et al., 2020; Li et al., 2021; Nagalapatti et al., 2022). Instead, we will provide a systematic study on the generalization performance of FL algorithms in relation to the Wasserstein distance.

## A.2  INTEGRATION OF FEDERATED LEARNING AND OPTIMAL TRANSPORT

Several studies have applied Optimal Transport (OT) to address the heterogeneity in FL frameworks. For example, (Reisizadeh et al., 2020) performs federated min-max optimization with OT to enhance robustness against distributional shifts. farnia2022optimal develops a personalized FL algorithm that learns OT mappings to align data points with a common distribution. nguyen2022generalization proposed a Wasserstein distributionally robust optimization algorithm, to handle all adversarial distributions inside the Wasserstein ball. Other research utilize Wasserstein distance as a metric to

assess data divergence across diverse domains. For example, tangfedimpro bounds the generalization performance by the conditional Wasserstein distance between data distributions of different clients. rakotomamonjyfederated proposes a method to calculate the Wasserstein distance in a federated manner. li2024data assesses data contribution in FL using the hierarchical Wasserstein distance.

### A.3 Data Valuation, Acquisition and Marketplace

Data valuation research focuses mainly on improving interpretability in machine learning by identifying the most influential, noisy, or misleading training examples, and could further help guide data selection and acquisition (Koh & Liang, 2017; Ghorbani & Zou, 2019; Jia et al., 2019; Pruthi et al., 2020; Just et al., 2023). In the data marketplace, data transactions can take two forms: one involves the transfer of data ownership (Lu et al., 2024), while the other involves transferring only the right to use the data, such as FL models (Zheng et al., 2022; Sun et al., 2024). This paper will focus on the later transaction form. Our work is also related to research predicting model performance associated with a particular data composition without performing actual training. For example, (Hashimoto, 2021) proposes the rational function to approximate the excess loss (the difference between the generalization error and the error of the best possible estimator). However, it only tackles the i.i.d setting, which might be impractical in real applications. It is also challenging to calculate the excess loss without knowing the oracle of the class. Furthermore, it could not help guide the selection of data for model training. Our work is closely related to (Kang et al., 2024), which utilizes the Wasserstein distance as a surrogate for the validation performance. However, it assumes that there are publicly available data from each data source, a condition that might restrict its applications when dealing with sensitive data. In contrast, we tackle a more challenging scenario where raw data sharing is not allowed, and the platform can't collect training data from multiple sources to **train a centralized model**. Instead, it is restricted to training a federated model. Notably, the non-i.i.d nature of the data in this federated setting poses significant challenges. Another concurrent work (Chhachhi & Teng, 2024) proposes a procurement mechanism for differentially private data based on the Wasserstein distance in the data marketplace, while our paper focus on the model training in the model marketplace and computes the Wasserstein distance between raw data as the performance surrogate. CLUES (Zhao et al.) identifies high-quality data from diverse private sources by monitoring per-sample gradients relative to both the private data and a public anchor dataset. This anchor dataset serves as a benchmark for evaluating the quality of candidate data. In contrast, our work addresses the scenario where this crucial anchor is the validation set from the data buyer, which should remain private.

## B Proof

### B.1 Proof of Theorem 4.1

First, we will prove the validation performance is bounded by the weighted average of the pair-wise Wasserstein distance, e.g. $\sum_{i=1}^{m} \alpha_i \mathcal{W}(D_i^{\mathrm{pi}}, D^{\mathrm{val}})$. We denote $f_t^i, f_v$ as the labeling function for training and validation data. Let $f(\theta)$ be the aggregated global model. Let $\{\mu_t^i\}_{i=1}^m, \mu_v$ be the training and validation distribution. Then we have

$$\mathbb{E}_{x \sim \mu_v(x)} \mathcal{L}\big(f_v(x), f(\theta, x)\big) - \mathcal{L}^{ERM}(\theta) = \mathbb{E}_{x \sim \mu_v(x)} \mathcal{L}\big(f_v(x), f(\theta, x)\big) - \sum_{i=1}^{m} \alpha_i \mathbb{E}_{x \sim \mu_t^i(x)} \mathcal{L}\big(f_t^i(x), f(\theta, x)\big)$$

$$= \sum_{i=1}^{m} \alpha_i \Big[ \mathbb{E}_{x \sim \mu_v(x)} \mathcal{L}\big(f_v(x), f(\theta, x)\big) - \mathbb{E}_{x \sim \mu_t^i(x)} \mathcal{L}\big(f_t^i(x), f(\theta, x)\big) \Big]$$

$$\leq \sum_{i=1}^{m} \alpha_i \Big[ \mathcal{W}(\mu_t^i, \mu_v) + \mathcal{O}(kM) \Big], \tag{11}$$

where the last inequality comes from the Theorem in (Just et al., 2023). However, this bound considers the quality and quantity of data available from each source individually, ignoring the relationships between sources. As another attempt, we provide a tighter bound when there exists a mixture of sources that approximates the target better than any single source, which is common in the non-i.i.d setting in the context of FL.

Based on the triangle inequality, we have

$$\left|\mathbb{E}_{x\sim\mu_v(x)}\mathcal{L}\big(f_v(x), f(\theta, x)\big) - \mathcal{L}^{ERM}(\theta)\right|$$

$$=\left|\mathbb{E}_{x\sim\mu_v(x)}\mathcal{L}\big(f_v(x), f(\theta, x)\big) - \sum_{i=1}^{m}\alpha_i\mathbb{E}_{x\sim\mu_t^i(x)}\mathcal{L}\big(f_t^i(x), f(\theta, x)\big)\right|$$

$$=\Big|\mathbb{E}_{x\sim\mu_v(x)}\mathcal{L}\big(f_v(x), f(\theta, x)\big) - \mathbb{E}_{x\sim\mu_v(x)}\mathcal{L}\big(f(\theta, x), f(\theta^\star, x)\big) + \mathbb{E}_{x\sim\mu_v(x)}\mathcal{L}\big(f(\theta, x), f(\theta^\star, x)\big)$$

$$+ \sum_{i=1}^{m}\alpha_i\mathbb{E}_{x\sim\mu_t^i(x)}\mathcal{L}\big(f(\theta, x), f(\theta^\star, x)\big) - \sum_{i=1}^{m}\alpha_i\mathbb{E}_{x\sim\mu_t^i(x)}\mathcal{L}\big(f(\theta, x), f(\theta^\star, x)\big) - \sum_{i=1}^{m}\alpha_i\mathbb{E}_{x\sim\mu_t^i(x)}\mathcal{L}\big(f_t^i(x), f(\theta, x)\big)\Big|$$

$$\leq \underbrace{\left|\mathbb{E}_{x\sim\mu_v(x)}\Big[\mathcal{L}\big(f_v(x), f(\theta, x)\big) - \mathcal{L}\big(f(\theta, x), f(\theta^\star, x)\big)\Big]\right|}_{U_1} + \underbrace{\left|\sum_{i=1}^{m}\alpha_i\mathbb{E}_{x\sim\mu_t^i(x)}\Big[\mathcal{L}\big(f(\theta, x), f(\theta^\star, x)\big) - \mathcal{L}\big(f_t^i(x), f(\theta, x)\big)\Big]\right|}_{U_2}$$

$$+ \underbrace{\left|\mathbb{E}_{x\sim\mu_v(x)}\mathcal{L}\big(f(\theta, x), f(\theta^\star, x)\big) - \sum_{i=1}^{m}\alpha_i\mathbb{E}_{x\sim\mu_t^i(x)}\mathcal{L}\big(f(\theta, x), f(\theta^\star, x)\big)\right|}_{U_3} \tag{12}$$

We further inspect the last term of the above inequality. We denote $D_\alpha$ the mixture of the $m$ source distributions with mixing weights equal to the components $\{\alpha_i\}_{i=1}^m$. Then we have the following result based on (Ben-David et al., 2010)

$$U_3 = \left|\mathbb{E}_{x\sim\mu_v(x)}\mathcal{L}\big(f(\theta, x), f(\theta^\star, x)\big) - \sum_{i=1}^{m}\mathbb{E}_{x\sim\mu_t^i(x)}\alpha_i\mathcal{L}\big(f(\theta, x), f(\theta^\star, x)\big)\right| \leq \mathcal{W}(\sum_{i=1}^{m}\alpha_i\mu_t^i, \mu_v) \tag{13}$$

$$U_1 = \left|\mathbb{E}_{x\sim\mu_v(x)}\Big[\mathcal{L}\big(f_v(x), f(\theta, x)\big) - \mathcal{L}\big(f(\theta, x), f(\theta^\star, x)\big)\Big]\right|$$

$$= \int\left|\mathcal{L}\big(f_v(x), f(\theta, x)\big) - \mathcal{L}\big(f(\theta, x), f(\theta^\star, x)\big)\right|d\mu_v(x) \leq k\int|f_v(x) - f(\theta^\star, x)|d\mu_v(x) \tag{14}$$

$$U_2 = \left|\sum_{i=1}^{m}\alpha_i\mathbb{E}_{x\sim\mu_t^i(x)}\Big[\mathcal{L}\big(f(\theta, x), f(\theta^\star, x)\big) - \mathcal{L}\big(f_t^i(x), f(\theta, x)\big)\Big]\right|$$

$$= \left|\sum_{i=1}^{m}\alpha_i\int\Big[\mathcal{L}\big(f(\theta, x), f(\theta^\star, x)\big) - \mathcal{L}\big(f_t^i(x), f(\theta, x)\big)\Big]d\mu_t^i(x)\right| \leq k\sum_{i=1}^{m}\alpha_i\int\left|f_t^i(x) - f(\theta^\star, x)\right|d\mu_t^i(x) \tag{15}$$

Therefore, we have

$$\left|\mathbb{E}_{x\sim\mu_v(x)}\mathcal{L}\big(f_v(x), f(\theta, x)\big) - \mathcal{L}^{ERM}(\theta)\right|$$

$$\leq \mathcal{W}(\sum_{i=1}^{m}\alpha_i\mu_t^i, \mu_v) + k\int\left|f_v(x) - f(\theta^\star, x)\right|d\mu_v(x) + k\sum_{i=1}^{m}\alpha_i\int\left|f_t^i(x) - f(\theta^\star, x)\right|d\mu_t^i(x) \tag{16}$$

### B.2 PROOF OF THEOREM 4.2

We provide the essential property B.1 from (Panaretos & Zemel, 2019) as follows

**Property B.1.** *For any vector $x \in \mathbb{R}^{d\times 1}$, $\mathcal{W}_2(X + x, Y + x) = \mathcal{W}_2(X, Y)$.*

We will begin our proof with the case of Gaussian distributions, as their Wasserstein distance has a clear analytical form, which could provide a rigorous approximation error bound. However, our theoretical analysis can be extended to more complex distributions.

Suppose $X_a \in \mathbb{R}^{m \times d} \sim \mathcal{N}(\mu_a, \sigma_a^2)$, $X_b \in \mathbb{R}^{n \times d} \sim \mathcal{N}(\mu_b, \sigma_b^2)$, $\gamma \in \mathbb{R}^{k \times d} \sim \mathcal{N}(\mu_\gamma, \sigma_\gamma^2)$. We consider 2-Wasserstein distance and the Kantorovich relaxation of mass splitting. Without loss of generality, we set $t = 0.5$. Then based on the barycentric mapping, the interpolating measures are

$$\eta_{X_a} = 0.5 \times X_a + 0.5 \times m[\pi(X_a, \gamma)\gamma],$$
$$\eta_{X_b} = 0.5 \times X_b + 0.5 \times n[\pi(X_b, \gamma)\gamma], \tag{17}$$

where $\pi(X_a, \gamma) \in \mathbb{R}^{m \times k}, \pi(X_b, \gamma) \in \mathbb{R}^{n \times k}$ are optimal transport plans.

(1) When $k = 1, \gamma = [\gamma_1, \cdots, \gamma_d]_{1 \times d}, \pi(X_a, \gamma) = [\frac{1}{m}]_{m \times 1}, \pi(X_b, \gamma) = [\frac{1}{n}]_{n \times 1}$, then based on Property B.1, $2\mathcal{W}_2(\eta_{X_a}, \eta_{X_b}) = \mathcal{W}_2(X_a + \gamma, X_b + \gamma) = \mathcal{W}_2(X_a, X_b)$

(2) When $k > 1$ and $k \neq m \neq n$. $\pi(X_a, \gamma) \in \mathbb{R}^{m \times k}, \pi(X_b, \gamma) \in \mathbb{R}^{n \times k}$. For $\pi(X_a, \gamma)$, we define $w_{i,l}$ as the value of the $(i, l)$-position value, where $i \in [1, m], l \in [1, d], w_i = \sum_{l=1}^{d} w_{i,l} = \frac{1}{m}$. Further, with uniform weights, there are $\lfloor \frac{m+k-1}{m} \rfloor$ non zero elements in each row of $\pi(X_a, \gamma)$. We denote the indices of the nonzero values in each row as the set $\mathcal{I}_i$. For simplicity, we assume all non-zero elements in $\pi(X_a, \gamma)$ has an uniform weight of $\frac{1}{m+k-1}$.

a. $k \to \infty$, then the weight is around $\frac{1}{k}$ if $l \in \mathcal{I}_i$ and 0 otherwise. In geometirc view, each point in $X_a$ are splited to map $k$ points in $\gamma$. Then we have

$$2\eta_{X_a} = X_a + m \times [\sum_{l=1}^{k} w_{i,l} \times \gamma_{l,j}]_{i,j=1}^{m,d} = \frac{m}{k} \times k[\mathbb{E}(\gamma_1), \cdots, \mathbb{E}(\gamma_d)]$$
$$= m[\bar{\gamma}_1, ..., \bar{\gamma}_d]_{1 \times d} \tag{18}$$

Then based on the Property B.1 we have $2\mathcal{W}_2(\eta_{X_a}, \eta_{X_b}) = \mathcal{W}_2(X_a, X_b)$.

b. When $k < \infty$, $2\eta_{X_a} = X_a + m \times [\sum_{l=1}^{k} w_{i,l} \times \gamma_{l,j}]_{i,j=1}^{m,d} = X_a + m \times \frac{1}{m+k-1}[\sum_{l=1}^{k} \mathbb{I}_{l \in \mathcal{I}_i} \gamma_{l,j}] = X_a + m \times \frac{1}{m+k-1} \times \frac{m+k-1}{m}[\bar{\gamma}_{i,j}^a]_{l,j=1}^{m,d} = X_a + [\bar{\gamma}_{i,j}^a]_{l,j=1}^{m,d}$. Similarly, $\eta_{X_b} = X_b + [\bar{\gamma}_{i,j}^b]_{i,j=1}^{n,d}$. If we denote $\bar{\gamma}^a = [\bar{\gamma}_{i,j}^a]_{l,j=1}^{m,d} = [\mu_\gamma + \sigma_a Z_a], \bar{\gamma}^b = [\mu_\gamma + \sigma_b Z_b]$, where $Z_a \in \mathbb{R}^{m \times d} \sim \mathcal{N}(0, 1), Z_b \in \mathbb{R}^{n \times d} \sim \mathcal{N}(0, 1)$, then

$$\sigma_a^2 = Var(\frac{m}{m+k-1} \sum_{l \in \mathcal{I}_i} \gamma_{l,j}) = [\frac{m}{m+k-1}]^2 Var(\sum_l \gamma_{l,j}). \tag{19}$$

As $\gamma_{l,j}$ is i.i.d sampled from $\mathcal{N}(\mu_\gamma, \sigma_\gamma^2)$, then $Var(\sum_l \gamma_{l,j}) = \sum_l Var(\gamma_{l,j}) = \sum_l \sigma_\gamma^2 = \frac{m+k-1}{m} \sigma_\gamma^2$. We can get $\sigma_a^2 = \frac{m}{m+k-1} \sigma_\gamma^2$. Similarly, $\sigma_b^2 = \frac{n}{n+k-1} \sigma_\gamma^2$

We define $p_a = \sqrt{\frac{m}{m+k-1}}, p_b = \sqrt{\frac{n}{n+k-1}}$. Therefore, our approximation is

$$2\mathcal{W}_2^2(\eta_{X_a}, \eta_{X_b}) = \mathcal{W}_2^2(X_a + p_a \sigma_\gamma Z_a, X_b + p_b \sigma_\gamma Z_b)$$
$$= \|\mu_a - \mu_b\|_2^2 + \|(\sigma_a^2 + p_a^2 \sigma_\gamma^2)^{\frac{1}{2}} - (\sigma_b^2 + p_b^2 \sigma_\gamma^2)^{\frac{1}{2}}\|_2^2. \tag{20}$$

Furthermore, we focus on the second term as

$$\|(\sigma_a^2 + p_a^2 \sigma_\gamma^2)^{\frac{1}{2}} - (\sigma_b^2 + p_b^2 \sigma_\gamma^2)^{\frac{1}{2}}\|_2^2$$

$$= (\sigma_a^2 + p_a^2 \sigma_\gamma^2) - (\sigma_b^2 + p_b^2 \sigma_\gamma^2) - 2\sqrt{(\sigma_a^2 + p_a^2 \sigma_\gamma^2)(\sigma_b^2 + p_b^2 \sigma_\gamma^2)}$$

$$= (\sigma_a^2 - \sigma_b^2) + \sigma_\gamma^2(p_a^2 - p_b^2) - 2 \underbrace{\sqrt{(\sigma_a \sigma_b)^2 + (\sigma_a p_b \sigma_\gamma)^2 + (p_a \sigma_\gamma \sigma_b)^2 + (p_a p_b \sigma_\gamma^2)^2}}_{K}$$

$$= \|\sigma_a - \sigma_b\|_2^2 + \sigma_\gamma^2(p_a - p_b)^2 + 2 \underbrace{(\sigma_a \sigma_b + p_a p_b \sigma_\gamma^2 - K)}_{H}$$

$$< \|\sigma_a - \sigma_b\|_2^2 + \sigma_\gamma^2(p_a - p_b)^2. \tag{21}$$

Then we can have an upper bound as

$$2\mathcal{W}_2^2(\eta_{X_a}, \eta_{X_b}) < \|\mu_a - \mu_b\|_2^2 + \|\sigma_a - \sigma_b\|_2^2 + \sigma_\gamma^2(p_a - p_b)^2 = \mathcal{W}_2^2(X_a, X_b) + \sigma_\gamma^2(p_a - p_b)^2 \tag{22}$$

Reversely,

$$H = \sigma_a \sigma_b + p_a p_b \sigma_\gamma^2 - \sqrt{(\sigma_a \sigma_b)^2 + (\sigma_a p_b \sigma_\gamma)^2 + (p_a \sigma_\gamma \sigma_b)^2 + (p_a p_b \sigma_\gamma^2)^2}$$

$$= \sqrt{(\sigma_a \sigma_b)^2} + \sqrt{(p_a p_b \sigma_\gamma^2)^2} - \sqrt{(\sigma_a \sigma_b)^2 + (\sigma_a p_b \sigma_\gamma)^2 + (p_a \sigma_\gamma \sigma_b)^2 + (p_a p_b \sigma_\gamma^2)^2}$$

$$> \sqrt{(\sigma_a \sigma_b)^2 + (p_a p_b \sigma_\gamma^2)^2} - \sqrt{(\sigma_a \sigma_b)^2 + (\sigma_a p_b \sigma_\gamma)^2 + (p_a \sigma_\gamma \sigma_b)^2 + (p_a p_b \sigma_\gamma^2)^2}$$

$$= \frac{(\sigma_a \sigma_b)^2 + (p_a p_b \sigma_\gamma^2)^2 - [(\sigma_a \sigma_b)^2 + (\sigma_a p_b \sigma_\gamma)^2 + (p_a \sigma_\gamma \sigma_b)^2 + (p_a p_b \sigma_\gamma^2)^2]}{\sqrt{(\sigma_a \sigma_b)^2 + (p_a p_b \sigma_\gamma^2)^2} + \sqrt{(\sigma_a \sigma_b)^2 + (\sigma_a p_b \sigma_\gamma)^2 + (p_a \sigma_\gamma \sigma_b)^2 + (p_a p_b \sigma_\gamma^2)^2}}$$

$$> -\sqrt{\frac{[(\sigma_a p_b \sigma_\gamma)^2 + (p_a \sigma_\gamma \sigma_b)^2]^2}{2(\sigma_a \sigma_b)^2 + 2(p_a p_b \sigma_\gamma^2)^2 + (\sigma_a p_b \sigma_\gamma)^2 + (p_a \sigma_\gamma \sigma_b)^2}} \tag{23}$$

Therefore, we have a lower bound

$$2\mathcal{W}_2^2(\eta_{X_a}, \eta_{X_b}) = \|\mu_a - \mu_b\|_2^2 + \|\sigma_a - \sigma_b\|_2^2 + \sigma_\gamma^2(p_a - p_b)^2 + 2H$$

$$> \mathcal{W}_2^2(X_a, X_b) + \sigma_\gamma^2(p_a - p_b)^2 - 2\underbrace{\sqrt{\frac{[(\sigma_a p_b \sigma_\gamma)^2 + (p_a \sigma_\gamma \sigma_b)^2]^2}{2(\sigma_a \sigma_b)^2 + 2(p_a p_b \sigma_\gamma^2)^2 + (\sigma_a p_b \sigma_\gamma)^2 + (p_a \sigma_\gamma \sigma_b)^2}}}_{M} \tag{24}$$

As for $M$, we will compare the value of the numerator and the denominator as

$$2(\sigma_a \sigma_b)^2 + 2(p_a p_b \sigma_\gamma^2)^2 + (\sigma_a p_b \sigma_\gamma)^2 + (p_a \sigma_\gamma \sigma_b)^2 - [(\sigma_a p_b \sigma_\gamma)^2 + (p_a \sigma_\gamma \sigma_b)^2]^2$$

$$= 2(\sigma_a \sigma_b)^2 + 2(p_a p_b)^2 \sigma_\gamma^4 + (p_b^2 + p_a^2)\sigma^2 \sigma_\gamma^2 - [(p_b^2 + p_a^2)^2(\sigma_a \sigma_b)^2]\sigma_\gamma^4$$

$$= (\sigma_a \sigma_b)^2[2 - (p_b^2 + p_a^2)^2 \sigma_\gamma^4] + 2(p_a p_b)^2 \sigma_\gamma^4 + (p_b^2 + p_a^2)\sigma^2 \sigma_\gamma^2, \tag{25}$$

then set $\sigma_\gamma^2 \leq \sqrt{\frac{2}{p_a^2 + p_b^2}}$ will definitely guarantee $0 < M < 1$.

Therefore, the approximation error $|2\mathcal{W}_2^2(\eta_{X_a}, \eta_{X_b}) - \mathcal{W}_2^2(X_a, X_b)|$ is bounded by $\sigma_\gamma^2(p_a - p_b)^2 \ll \sigma_\gamma^2$. When $p_a = p_b$ or $k \to \infty$, we have $2\mathcal{W}_2^2(\eta_{X_a}, \eta_{X_b}) = \mathcal{W}_2^2(X_a, X_b)$.

Overall, the approximation gap is affected only by $\sigma_\gamma$ and $k$. Specifically, given a larger $k$, $(p_a - p_b)^2$ becomes smaller, resulting in a better estimation.

## C ADDITIONAL EXPERIMENTS

### C.1 EFFECTIVENESS OF PRIVATE WASSERSTEIN DISTANCE

We aim to conduct the comparison between our proposed method and the previous approximation approach, FedWad, in terms of estimation error and computational time. The ground truth for our analysis is obtained through the direct calculation of the Wasserstein distance using raw data. For data processing, we randomly sample $\mathbf{x}_1^\mu$ and $\mathbf{x}^\nu$ with equal sizes and their distributions do not necessarily to be identical. Then we mislabel $20\%$ of data points in $\mathbf{x}_1^\mu$ and construct the $\mathbf{x}_2^\mu$. Our comparison is carried out in two main scenarios: First, we focus on the Wasserstein distance between two parties, where $\mathbf{x}_1^\mu$ and $\mathbf{x}_2^\mu$ is stored by one party, $\mathbf{x}^\nu$ is stored by the other party. Second, we compute the distance $\mathcal{W}(\sum \mathbf{x}_i^\mu, \mathbf{x}^\nu)$ among multiple sources, where $\{\mathbf{x}_i^\mu\}_{i=1}^3$ are stored across three different sources. As presented in Table 2,despite having access to the same information as FedWad for performing approximations, our method not only maintains a competitively low estimation error but also achieves a markedly higher computational efficiency, demonstrating it adaptability in different scenarios.

### C.2 WASSERSTEIN DISTANCE AS A SURROGATE FOR THE VALIDATION PERFORMANCE IN THE I.I.D SETTING

We conducted our experiment in an i.i.d. setting using the CIFAR10 dataset. From the training set, we randomly select 6,000 data points and divide them into three equal parts, each containing 2,000 data points. These parts represent separate local datasets in our federated learning setup.

|  | Ground truth | FedWad | Ours |
|---|---|---|---|
| Metrics | CIFAR10 | | |
| $\mathcal{W}(\mathbf{x}_1^\mu, \mathbf{x}^\nu)$ | 27.46 | 32.90 | 32.88 |
| $\mathcal{W}(\mathbf{x}_2^\mu, \mathbf{x}^\nu)$ | 571.73 | 571.99 | 572.01 |
| $\mathcal{W}(\sum \mathbf{x}_i^\mu, \mathbf{x}^\nu)$ | 487.72 | NA | 488.44 |
| | Fashion | | |
| $\mathcal{W}(\mathbf{x}_1^\mu, \mathbf{x}^\nu)$ | 12.68 | 15.59 | 15.67 |
| $\mathcal{W}(\mathbf{x}_2^\mu, \mathbf{x}^\nu)$ | 295.17 | 295.29 | 296.38 |
| $\mathcal{W}(\sum \mathbf{x}_i^\mu, \mathbf{x}^\nu)$ | 687.69 | NA | 688.94 |
| Avg.time | - | 2.55 | **0.17** |

Table 2: Our method obtains similar approximations of Wasserstein distance while requiring less computation time."NA" means FedWad cannot be applied to multi-source scenarios.

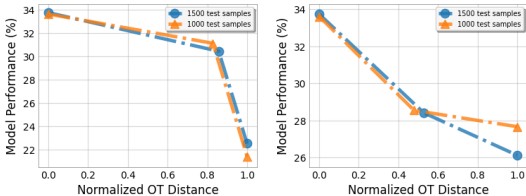

Figure 7: Model performance vs. Combined Wasserstein Distance for i.i.d data

In the first setting, we introduce varying levels $\epsilon = \{0, 1, 5\}$ of label noise to the three local datasets. Noise levels simulate degrees of data corruption, and higher values represent more significant distortions to the data.

In the second setting, we explore the effect of imbalanced label distributions. The data splitting is as follows: (1) Clean data, with each label evenly distributed across the dataset. (2) Imbalanced data with classes $0, 1, 2, 3$ (major classes) having a combined proportion of $70\%$, and the remaining classes distributed uniformly across the remaining $30\%$. (3) Highly imbalanced data with classes $0, 1, 2, 3$ (major classes) having a combined proportion of $91\%$, and the remaining classes distributed uniformly across the remaining $9\%$.

Each local dataset is trained with its corresponding noise level, and the global model is aggregated using the FedProx framework to mitigate data heterogeneity. To quantify the degree of data distribution alignment, we compute the combined Wasserstein distance between local datasets. For improved interpretability, the Wasserstein distances are normalized through min-max scaling. Model performance is evaluated across varying noise levels using standard metrics including classification accuracy and cross-entropy loss. We analyze the correlation between the normalized Wasserstein distance and the global model's performance to investigate the impact of data heterogeneity, as illustrated in Figure 7.

## C.3    DATA SELECTION FOR UNLABELED DATA

We explore a more challenging setting, where the model buyer only has unlabeled test data. Specifically, given a set of unlabeled test data $D^{\text{test}} = \{x_1^{\text{test}}, \cdots, x_m^{\text{test}}\}$, the selection task is to select valuable subsets of training data from each source, so that the trained model will have a smaller prediction loss on the test data. Following a similar experimental setup as in (Lu et al., 2024), we conduct experiments on one real-world medical dataset: the RSNA Pediatric Bone Age dataset, where the task is to assess bone age (in months) from X-ray images. To extract features, each image is embedded using a CLIP ViT-B/32 model (Radford et al., 2021). As there is no label information in our setting, we can not train federated models and evaluate the model performance. Therefore, our selection procedure is model-agnostic in this setting.

For single-source scenarios, we employ the gradient score from Equation equation 5 for data selection. With $|D^{\text{train}}| = 1000$ and $|D^{\text{test}}| = 50$, we perform training data selection under varying budgets. The seller computes the interpolating measure $\eta_{\mathbf{x}^{\text{train}}}$ using features $\mathbf{x}^{\text{train}}$, and the buyer calculates the interpolating measure $\eta_{\mathbf{x}^{\text{test}}}$.These measures are then used as inputs to compute the calibrated score.

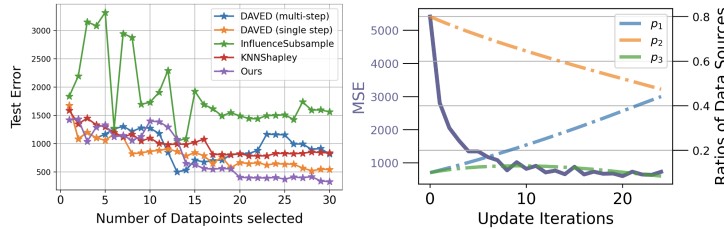

Figure 8: Experiments were conducted in both single and multi-source scenarios. In single-source cases (left), our method achieves lower MSE compared to others. For multi-source scenarios (right), our approach determines the optimal mixing ratio, leading to consistently decreasing MSE.

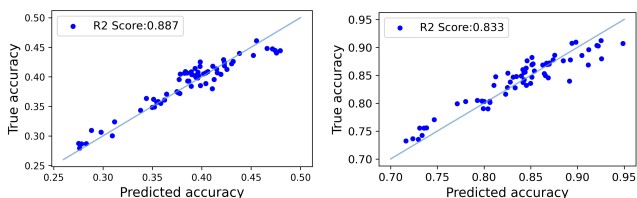

Figure 9: Predicted Accuracy vs. True Accuracy for unseen data scales on CIFAR and MNIST.

After optimization, we select the top-$k$ most valuable data points (those with the largest negative gradient scores) and train a regression model to predict the test data. We compare our approach with other baselines on the test mean squared error (MSE). As demonstrated in Figure 8 (left panel), our selection algorithm achieves superior performance, yielding lower test MSE compared to baseline methods.

In multi-source scenarios with three sellers and one buyer, we address the mixing ratio optimization problem. In this setting, we also consider three data sellers and one model buyer. The buyer has 300 test data points, covering 7 different labels, and the ground-truth distribution of labels is non-i.i.d. Each data seller has non-overlapping labels with others, and the label distribution is also non-i.i.d. As the iteration procedure requires $\frac{\partial \hat{V}(D(N,\mathbf{p}),D^{\text{val}})}{\partial \mathbf{p}}$, which is not applicable when there is only unlabeled test data, we use $\frac{\partial \mathcal{W}(D(N,\mathbf{p}),D^{\text{val}})}{\partial \mathbf{p}}$ as the gradient, which is easily available. As shown in Figure 8 (right side), the MSE continuously decreases during the iterations for optimizing the mixing ratio.

## D PERFORMANCE ESTIMATORS

### D.1 CHOICES OF THE PERFORMANCE ESTIMATORS

Suppose the budget of the trial runs is $B_s$. In the $j$-th trial run, the platform randomly samples the mixing ratio $\mathbf{p}^j$ and the data budget $N_j$, where each $\mathbf{p}^j = \{p_i^j\}_{i=1}^m$ is sampled from a probability simplex, and $N_j \in \{1, \cdots, \sum_{i=1}^m |D_i^{\text{pi}}|\}$, Therefore, the $i$-th data seller only utilizes the subset $S_i^j \subseteq D_i^{\text{pi}}, |S_i^j| = p_i^j N_j$ to conduct local training. We denote the constructed training data is $D(N_j, \mathbf{p}^j) = \sum_{i=1}^m S_i^j$. The platform approximates the Wasserstein distance $\mathcal{W}(D(N_j, \mathbf{p}^j), D^{\text{val}})$, operates the federated training and gets the validation performance $V(D(N_j, \mathbf{p}^j), D^{\text{val}})$ from the model buyer. Upon finishing the trial runs, the set of tuples $\left\{\mathbf{p}^j, N_j, \mathcal{W}(D(N_j, \mathbf{p}^j), D^{\text{val}}), V(D(N_j, \mathbf{p}^j), D^{\text{val}})\right\}_{j=1}^{B_s}$ are collected, which could be further leveraged to construct the performance estimator as $\hat{V}_j(D(N_j, \mathbf{p}^j), D^{\text{val}}) = f_j(\mathbf{p}^j, \mathcal{W}(D(N_j, \mathbf{p}^j), D^{\text{val}}))$. We incorporate the mixing ratio because the model behavior in FL is sensitive to the number of contributing groups and the weight of each local model. The potential choices of the performance estimator are discussed in Appendix D.1.

In our analysis in Sec 4.1, we observe an inverse correlation between the combined Wasserstein distance and validation performance upon model convergence, and theoretically prove that the model performance is bounded by an affine transformation of this distance. Consequently, we first define a baseline estimator as:

$$f(D(N, \mathbf{p}), D^{\text{val}}) = a_1 \mathcal{W}(D(N, \mathbf{p}), D^{\text{val}}) + a_0 \tag{26}$$

where $a_1$ and $a_0$ are learnable parameters. However, in federated learning scenarios where client participation dynamically scales (via additions/removals), the mixing ratio $\mathbf{p}$ exerts a pronounced influence: assigning negligible weights ($p_i \approx 0$) or excluding data sources ($p_i = 0$) directly impacts model behavior by altering the effective number of contributing groups. Therefore, based on (Kang et al., 2024), an enhanced estimator incorporates $\mathbf{p}$ to address this heterogeneity

$$V(D(N, \mathbf{p}), D^{\text{val}}) = \sum_{i=1}^{m}(b_2^i \cdot p_i^2 + b_1^i \cdot p_i + b_0)\mathcal{W}(D(N, \mathbf{p}), D^{\text{val}}) + \sum_{i=1}^{m} c_1^i \cdot p_i. \tag{27}$$

Furthermore, we could approximate the change of the model performance.

$$\hat{V}_j(D(N_j, \mathbf{p}^j), D^{\text{val}}) = V_i(D(N_i, \mathbf{p}^i), D^{\text{val}}) + f(\triangle \mathbf{p}, \triangle \mathcal{W}) \tag{28}$$

These three estimators are sufficient to provide reliable performance predictions in most circumstances.

However, it is sensitive to the convergence of an FL model. For example, if the model fails to combine local information in a high heterogeneous setting, and is prone to one local model, the representation ability is poor. We conjecture the combined Wasserstein distance could be a signal to determine whether the model is convergence. More discussions are in Appendix E.

## D.2 Baselines of Performance Estimators

**Linear:** Assume a functional form of $\hat{V}(D(N, \mathbf{p}), D^{\text{val}}) = a \log(N) + \mathbf{b}^T \mathbf{q} + c$, where $\mathbf{b}^T = \{b_0, b_1, \cdots, b_m\}$

**Pseudo-Quadratic:** $\hat{V}(D(N, \mathbf{p}), D^{\text{val}}) = \sum_{i=1}^{m}(c_2^i \cdot p_i^2 + c_1^i \cdot p_i + c_0) + b \log(N)$

**Quadratic:** $\hat{V}(D(N, \mathbf{p}), D^{\text{val}}) = \sum_{i=1}^{m}(c_2^i \cdot p_i^2 + c_1^i \cdot p_i + c_0) + \sum_{i=1}^{m}\sum_{j=1}^{i}(c_3^{ij} \cdot p_i p_j) + b \log(N)$

**Rational:** $\hat{V}(D(N, \mathbf{p}), D^{\text{val}}) = \sum_{i=1}^{m}\left(\sum_{j=1}^{m} c^{ij} \cdot p_j\right)^{-1} + b \log(N)$

**AggWad:** $\hat{V}(D(N, \mathbf{p}), D^{\text{val}}) = a \times \left[\sum_{i=1}^{m} \alpha_i \mathcal{W}(S_i, D^{\text{val}})\right] + \mathbf{b}^T \mathbf{q} + c$, where $\alpha_i$ is the aggregation weight in FL as in equation 10.

## E Additional Discussions

This section examines the critical relationship between FL algorithms and the efficacy of the combined Wasserstein distance as both a convergence indicator and a data selection metric. The combined Wasserstein distance reliably serves as a surrogate for validation performance only when global models achieve stable convergence (i.e., low training loss and high validation accuracy). However, its sensitivity to FL algorithm performance necessitates careful interpretation when used for data selection.

To validate this hypothesis, we conduct a comparative experiment using FedAvg and FedProx under extreme non-i.i.d. data distributions. For FedProx, we set two different levels of regularizations to control how far the local model from the global model. A larger number indicates a larger penalty. We aim to demonstrate that algorithmic choices fundamentally influence the interpretability and utility of the combined Wasserstein distance. For all algorithms, we set local epochs to 10 and global iterations to 80. For FedProx, we consider two different regularizations: 0.1 and 0.3. The validation performance is recorded every 10 iterations, and the results are visualized in Figure 10, Figure 11,Figure 12. Green, orange, and blue dots represent models trained with three, two, and one data source, respectively. We normalize the Wasserstein distance.

There are several observations and insights. First, FedProx(0.1) and FedProx(0.3) achieve significantly better validation accuracy (FedProx(0.1) achieves nearly 42%, FedProx(0.3) achieves nearly 46%)

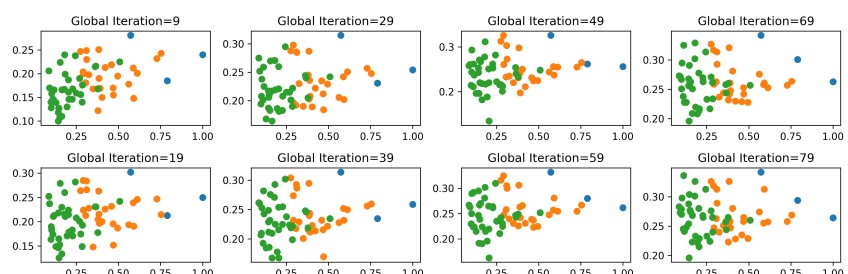

Figure 10: Model performance vs. Combined Wasserstein Distance with FedAvg

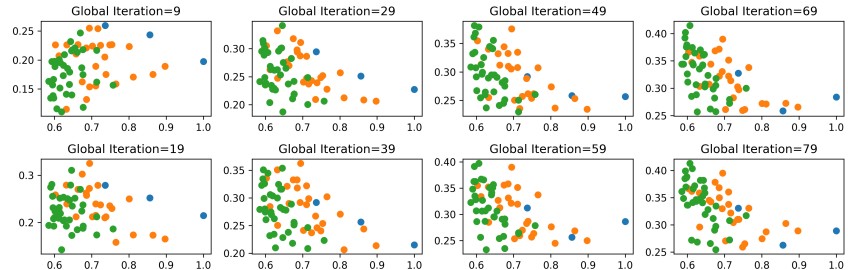

Figure 11: Model performance vs. Combined Wasserstein Distance with FedProx(0.1)

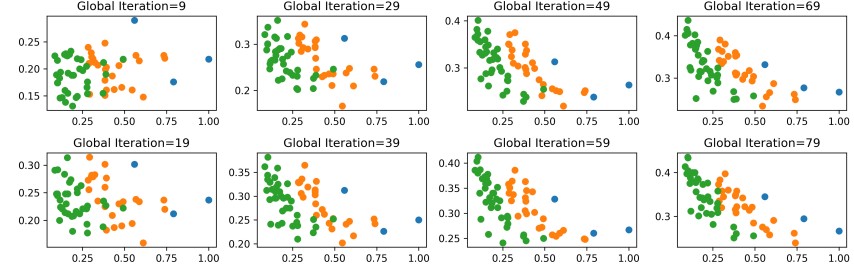

Figure 12: Model performance vs. Combined Wasserstein Distance with FedProx(0.3)

compared to FedAvg (35%) by the 79th global iteration. Second, in FedProx, models trained on three sources consistently outperform those trained on two or one source. This aligns with the ground truth, as each source contains only partial labels, while the validation set is balanced and covers all labels. However, in FedAvg, models trained on two sources have competitive performance with three sources. Third, despite minor fluctuations, a smaller Wasserstein distance generally correlates with better validation performance in FedProx. In contrast, FedAvg exhibits no such trend, indicating that the Wasserstein distance loses its representational utility when the model fails to converge. These findings underscore the importance of algorithmic stability in leveraging the combined Wasserstein distance for effective data selection and convergence monitoring.

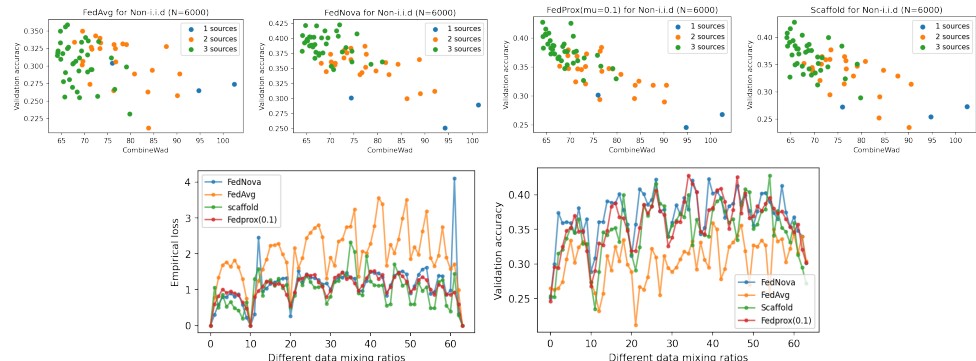

Figure 13: A comparison of FedAvg, FedNova, FedProx and Scaffold in a three-source setting ($N = 6000$). FL models with lower training loss and better validation performance has a more distinct correlation between validation performance and CombineWad.

# F    QUANTITATIVE LOWER BOUND OF RECONSTRUCTION RISK

We now formalize the privacy guarantees of our construction in a Bayesian sense. Let the private dataset be a matrix $P \in \mathbb{R}^{m \times d}$, representing $m$ data points $\mathbf{x}_i \in \mathbb{R}^d$. Our mechanism $M(P)$ produces an obfuscated matrix $\mathbf{Z} \in \mathbb{R}^{m \times d}$ via interpolation with a random reference measure $\gamma$.

**Definition F.1** (Geometric Obfuscation Mechanism). Let $\gamma$ be a reference measure composed of $k$ support points $\mathbf{x}^\gamma \in \mathbb{R}^{k \times d}$, sampled i.i.d. from $\mathcal{N}(\mu_\gamma, \sigma_\gamma^2 I_d)$. The mechanism $M(P)$ proceeds as follows:

1. Compute the (regularized) optimal transport (OT) plan $\mathbf{P}^\star(P, \gamma) \in \mathbb{R}^{m \times k}$ between the empirical distribution of $P$ and the reference measure $\gamma$.

2. Compute the barycentric map $\mathbf{Y}(P, \gamma) \in \mathbb{R}^{m \times d}$, whose $i$-th row is

$$\mathbf{Y}_i(P, \gamma) = \left( m \, \mathbf{P}^\star(P, \gamma) \, \mathbf{x}^\gamma \right)_i = \sum_{j=1}^k w_{ij}(P, \gamma) \, \mathbf{x}_j^\gamma,$$

where the weights $w_{ij}(P, \gamma)$ satisfy $\sum_{j=1}^k w_{ij}(P, \gamma) = 1$.

3. For an interpolation parameter $t \in [0, 1]$, the output matrix $\mathbf{Z} = M(P)$ has rows

$$\mathbf{z}_i(t) = (1 - t) \, \mathbf{x}_i + t \, \mathbf{Y}_i(P, \gamma), \qquad i = 1, \ldots, m.$$

We measure privacy through the lens of *reconstruction risk*, i.e., the minimum mean-squared error (MMSE) attainable by any adversary who observes $\mathbf{Z}$ and attempts to reconstruct the original $P$.

**Definition F.2** (Reconstruction Risk $R$). The reconstruction risk $R$ is defined as

$$R = \inf_{\hat{P}} \mathbb{E}_{P, \gamma} \left[ \| P - \hat{P}(\mathbf{Z}) \|_F^2 \right],$$

where the infimum is taken over all measurable estimators $\hat{P} : \mathbb{R}^{m \times d} \to \mathbb{R}^{m \times d}$, and the expectation is with respect to both the prior on $P$ and the randomness of the reference measure $\gamma$. A larger value of $R$ corresponds to stronger privacy.

We next derive a quantitative lower bound on $R$. The key idea is to (i) relate $R$ to the mutual information between $P$ and $\mathbf{Z}$ via entropy–MMSE inequalities, and (ii) upper-bound this information leakage by analyzing the conditional distribution of $\mathbf{Z}$ given $P$ under a Gaussian approximation.

**Theorem F.3** (Reconstruction Privacy Bound). *Let $n = md$ be the dimension of the vectorized representations of $P$ and $\mathbf{Z}$. Assume that, for each fixed $P$, the conditional distribution $p(\mathbf{Z}|P)$ can be approximated by a multivariate Gaussian $\mathcal{N}\big(\mu_{\mathbf{Z}}(P), \Sigma_{\mathbf{Z}}(P)\big)$, and that the conditional*

*covariance $\Sigma_{\mathbf{Z}}(P)$ does not vary dramatically across typical datasets $P$ (so that its trace can be treated as a dataset-level quantity). Then the reconstruction risk satisfies*

$$R \geq f\big(\mathrm{Tr}\big(\Sigma_{\mathbf{Z}}(P)\big)\big),$$

*for some non-decreasing function $f$. Moreover, under the Gaussian reference measure in Definition F.1 and the barycentric interpolation above, the total conditional variance admits the approximation*

$$\mathrm{Tr}\big(\Sigma_{\mathbf{Z}}(P)\big) \approx \frac{t^2\, m\, d\, \sigma_\gamma^2}{k_{\text{eff}}(\varepsilon)},$$

*where*

$$k_{\text{eff}}(\varepsilon) := \left(\frac{1}{m}\sum_{i=1}^{m}\sum_{j=1}^{k} w_{ij}(P,\gamma)^2\right)^{-1}$$

*is the effective support size of the Sinkhorn-smoothed OT weights $w_{ij}(P,\gamma)$ with entropic regularization parameter $\varepsilon$. Under mild homogeneity assumptions on these weights, $k_{\text{eff}}(\varepsilon)$ concentrates and can be treated as a dataset-level quantity. Consequently,*

$$R \gtrsim f\left(\frac{t^2\, m\, d\, \sigma_\gamma^2}{k_{\text{eff}}(\varepsilon)}\right),$$

*showing that reconstruction privacy improves with the total conditional variance induced by the random reference measure.*

*Proof.* We first relate the reconstruction risk to mutual information. Let $\mathbf{X} = \mathrm{vec}(P) \in \mathbb{R}^n$ and $\mathbf{Z} = \mathrm{vec}(M(P)) \in \mathbb{R}^n$. The optimal estimator in the MMSE sense is the conditional mean, and thus

$$R = \inf_{\hat{P}} \mathbb{E}\|P - \hat{P}(\mathbf{Z})\|_F^2 = \mathbb{E}_{\mathbf{Z}}\big[\mathrm{Tr}\big(\mathrm{Cov}(\mathbf{X}|\mathbf{Z})\big)\big]. \tag{29}$$

Classical entropy–covariance inequalities imply that, for each realization $\mathbf{Z} = z$, the conditional covariance $\Sigma_{\mathbf{X}|\mathbf{Z}}(z)$ satisfies

$$\mathrm{Tr}\big(\Sigma_{\mathbf{X}|\mathbf{Z}}(z)\big) \geq \frac{n}{2\pi e}\exp\left(\frac{2}{n}h(\mathbf{X}|\mathbf{Z}=z)\right),$$

where $h(\cdot)$ denotes differential entropy. Taking expectations over $z$ and using Jensen's inequality yields a lower bound of the form

$$R \geq g\big(I(\mathbf{X};\mathbf{Z})\big),$$

for some decreasing function $g$, where $I(\mathbf{X};\mathbf{Z})$ denotes mutual information. Intuitively, the *larger* the conditional uncertainty $h(\mathbf{X}|\mathbf{Z})$, the *smaller* the information leakage, and the *larger* the reconstruction risk $R$.

We now quantify this conditional uncertainty by analyzing the channel $p(\mathbf{Z}|P)$ induced by the random reference measure $\gamma$. For each $i \in \{1,\dots,m\}$,

$$\mathbf{z}_i(t) = (1-t)\,\mathbf{x}_i + t\,\mathbf{Y}_i(P,\gamma), \qquad \mathbf{Y}_i(P,\gamma) = \sum_{j=1}^{k} w_{ij}(P,\gamma)\,\mathbf{x}_j^\gamma,$$

where the weights $w_{ij}(P,\gamma)$ are determined by the OT plan and satisfy $\sum_{j=1}^{k} w_{ij}(P,\gamma) = 1$. Conditioned on $P$, the only source of randomness is the reference points $\mathbf{x}_j^\gamma \sim \mathcal{N}(\mu_\gamma, \sigma_\gamma^2 I_d)$, which we assume yield an approximately Gaussian channel

$$\mathbf{Z}|P \approx \mathcal{N}\big(\mu_{\mathbf{Z}}(P), \Sigma_{\mathbf{Z}}(P)\big).$$

The conditional mean is

$$\mu_{\mathbf{Z}}(P)_i = \mathbb{E}_\gamma[\mathbf{z}_i(t)|P] = (1-t)\,\mathbf{x}_i + t\,\mathbb{E}_\gamma[\mathbf{Y}_i(P,\gamma)|P],$$

and the block $(i, j)$ of the conditional covariance matrix is

$$\Sigma_{\mathbf{Z}}(P)_{ij} = \mathrm{Cov}_\gamma(\mathbf{z}_i(t), \mathbf{z}_j(t)|P) = t^2 \, \mathrm{Cov}_\gamma(\mathbf{Y}_i(P, \gamma), \mathbf{Y}_j(P, \gamma)|P),$$

since $(1 - t)\mathbf{x}_i$ is deterministic given $P$. In particular, the total conditional variance is

$$\mathrm{Tr}\left(\Sigma_{\mathbf{Z}}(P)\right) = \sum_{i=1}^{m} \mathrm{Tr}\left(\Sigma_{\mathbf{Z}}(P)_{ii}\right)$$

$$= \sum_{i=1}^{m} \mathrm{Tr}(\mathrm{Var}_\gamma(\mathbf{z}_i(t)|P))$$

$$= t^2 \sum_{i=1}^{m} \mathrm{Tr}(\mathrm{Var}_\gamma(\mathbf{Y}_i(P, \gamma)|P)).$$

Using the linear representation of $\mathbf{Y}_i(P, \gamma)$ and the independence of the Gaussian reference points, we obtain

$$\mathrm{Var}_\gamma(\mathbf{Y}_i(P, \gamma)|P) = \sum_{j=1}^{k} w_{ij}(P, \gamma)^2 \, \mathrm{Var}_\gamma(\mathbf{x}_j^\gamma) = \sigma_\gamma^2 \Big( \sum_{j=1}^{k} w_{ij}(P, \gamma)^2 \Big) I_d,$$

and hence

$$\mathrm{Tr}(\mathrm{Var}_\gamma(\mathbf{Y}_i(P, \gamma)|P)) = d \, \sigma_\gamma^2 \sum_{j=1}^{k} w_{ij}(P, \gamma)^2.$$

Substituting back yields

$$\mathrm{Tr}\left(\Sigma_{\mathbf{Z}}(P)\right) = t^2 d\sigma_\gamma^2 \sum_{i=1}^{m} \sum_{j=1}^{k} w_{ij}(P, \gamma)^2.$$

Under mild homogeneity assumptions on the Sinkhorn-smoothed OT weights $w_{ij}(P, \gamma)$ (e.g., approximate row-wise concentration for a fixed regularization parameter $\varepsilon$), we define the effective support size

$$k_{\mathrm{eff}}(\varepsilon) := \left( \frac{1}{m} \sum_{i=1}^{m} \sum_{j=1}^{k} w_{ij}(P, \gamma)^2 \right)^{-1},$$

and treat it as approximately dataset-level. This gives the approximation

$$\mathrm{Tr}\left(\Sigma_{\mathbf{Z}}(P)\right) \approx \frac{t^2 m d \sigma_\gamma^2}{k_{\mathrm{eff}}(\varepsilon)}.$$

Finally, under the Gaussian channel approximation, the mutual information $I(P; \mathbf{Z})$ is a decreasing function of the conditional covariance $\Sigma_{\mathbf{Z}}(P)$ (equivalently, of its trace). Combining this with the entropy–MMSE argument above shows that $R$ is lower bounded by a non-decreasing function of $\mathrm{Tr}(\Sigma_{\mathbf{Z}}(P))$, and hence by a non-decreasing function of $t^2 m d \sigma_\gamma^2 / k_{\mathrm{eff}}(\varepsilon)$. This establishes the claimed bound. $\qquad\square$

