# OpenReview forum: "Data Valuation and Selection in a Federated Model Marketplace"
_ICLR.cc/2026/Conference — Submitted to ICLR 2026_

### Official Review · Reviewer_HUeP · 2025-10-31

**Soundness:** 2
**Presentation:** 2
**Contribution:** 2
**Rating:** 4
**Confidence:** 3

**Summary:**

This paper proposes a new data valuation method based on private Wasserstein distance estimation that is suitable for a decentralized data market. This setting is novel and realistic in that data buyers request an FL model trained on a combination of inaccessible and heterogeneous data sources, while data owners bid by selecting a portion of their data to be used for training.

**Strengths:**

- It is intriguing to expand the usage of Wasserstein distance, as a credible model performance predictor, into a non-IID, data-decentralized setting.
- The proposed CombineWad method is theoretically and empirically supported.
- In practice, the target problem setting itself is critical and well-aligned with the needs of data-centric FL. It is impossible to train FL models for all combinations of private datasets; thus, preliminary dataset curation should be considered before operating FL.

**Weaknesses:**

- The existence of trusted central platform other than a model buyer can be a strict assumption in practice.
- The empirical validation is not sufficient overall to prove the efficacy of the proposed method.
  - The number of clients is only limited to three, which is far from realistic setting. Either two options below could possibly resolve this issue:
    - 1) Please consider limiting the target FL setting into a cross-silo FL setup where moderate number of credible clients are participating, to hopefully persuade the small number of clients in the experiments.
    - 2) Or, please consider providing additional results for proving the scalability of the proposed method using widely-used FL benchmark having sufficient number of clients, e.g., `FEMNIST` (3K+ clients), `Sent140` (600K+ clients), and `StackOverflow` (300K+ clients).
  - The aggregation algorithms should have been more diverse and up-to-date, including state-of-the-art adaptive aggregation algorithms, e.g., `TERM` and `AAggFF`. This is essential in order to conclusively demonstrate that the 'Technical Challenge 2' (line 89) has been resolved.

- `FEMNIST` & `Sent140`: https://arxiv.org/abs/1812.01097
- `StackOverflow`: https://www.tensorflow.org/federated/api_docs/python/tff/simulation/datasets/stackoverflow/load_data
- `TERM`: https://arxiv.org/abs/2007.01162
- `AAggFF`: https://arxiv.org/abs/2405.20821

**Questions:**

- The existence of validation data for calculating Wasserstein distance should have been more convincing. For example, the validation dataset itself has too limited a diversity of sample sizes to allow the model buyers to train an independent model.
  - It seems that $k$ in Theorem 4.2 should be in similar magnitude with the number of pilot ($n$) and validation dataset ($m$). In practice, this can incur an unavoidable communication cost, if $n$ and $m$ are large. Any guide to set $k$ in Theorem 4.2, and what values are used throughout experiments?
- Just to confirm, does Figure 1 imply that the utility of pilot data is maintained linearly, proportionally to the training budget?
- Could the authors please clarify the "single-seller limitations" in line 294?
- Setting $t=0$ for Eq. (6) coincides with the empirical distribution of the pilot dataset. Doesn't this result in privacy leakage?
- Does the Eq. (9) guarantee convergence to global optimum, i.e., is Eq. (8) convex in $\mathbf{p}$? Any reference is appreciated.
- Why there exist periodic fluctuations in validation loss and accuracy in Figure 4 and 13?
- Which dataset do the results in Figure 4 correspond to?
- With the existence of va, is there sharing of pilot data using randomized SVD?
- Please consider using principled label-noise simulation methods for the "Data Selection with Mislabeled Data" experiments in line 469, such as the pairwise flipping or symmetric flipping.

* Pairwise Flipping: https://arxiv.org/abs/1804.06872
* Symmetric Flipping: https://arxiv.org/abs/1505.07634

### Minor comments
- Please consider changing citation style in the draft, e.g., (Alphabet et al., 2021), for better readability.
- Missing citation: "The Monge problem" in line 129; "The Kantorovich's relaxed formulation" in line 132; all benchmark datasets except the last one.
- Typo: "Table ??" in line 161; "as reported in ?" in line 292; "combineWad -> CombineWad' in line 280, 284, and 331.
- Please add a legend in Figure 1.
- $\alpha$ is already used in Theorem 4.1 Please consider changing the step size notation in Eq. (9) to others to avoid any confusion.
- Please consider increasing font sizes and DPI in Figure 4.

---

> ### Author Response · Authors · 2025-12-03
> **Response to Reviewer HUeP [1]**
>
> Dear Reviewer HUeP,
>
> Thanks for your constructive feedback. Hope our response could address your concerns.
>
> **Q1.The existence of trusted central platform other than a model buyer can be a strict assumption in practice.**
>
> We acknowledge this concern and offer the following clarifications: 1. We assume an "Honest-but-Curious" (semi-trusted) model, not full trust. The platform facilitates orchestration but is technically restricted from accessing raw data by our privacy mechanism. 2.A central orchestrator is standard in real-world marketplaces (e.g., FATE) to minimize communication complexity.  3. Crucially, our method is not strictly bound to a central platform. Since our obfuscated proxy $\mathbf{Z}$ guarantees privacy (as detailed in General Response 2), clients can theoretically broadcast $\mathbf{Z}$ in a Peer-to-Peer (P2P) fashion for valuation. This validates that the central platform is an efficiency choice (avoiding $O(m^2)$ communication) rather than a strict theoretical constraint.
>
> **Q2.Just to confirm, does Figure 1 imply that the utility of pilot data is maintained linearly, proportionally to the training budget?**
>
> No. Figure 1 illustrates ranking consistency, not a linear relationship. The figure demonstrates that data mixtures achieving high accuracy on small budgets consistently maintain their superiority as the data scale increases. This stability of optimality confirms that we can reliably identify the optimal data mixture using small-scale pilot data and then project its performance to larger budgets.
>
> **Q3.Could the authors please clarify the "single-seller limitations" in line 294? **
>
> The phrase "single-seller limitations" refers to the inherent inability of prior Federated Wasserstein estimators (e.g., computing pairwise distances) to capture the pairwise value of data sources. However, in non-IID settings, individual sellers often possess partial or label-skewed data. Evaluating them in isolation fails to account for data complementarity, where two data sources might have high distances individually but form a high-quality, balanced distribution when combined.
>
> In contrast, we propose a new approximation algorithm (with a total different optimization technique) to evaluate the Wasserstein distance between the aggregated mixture distribution of multiple sources and the validation set. This allows us to optimize the mixing ratios ($\mathbf{p}$) to exploit interactions between sources, enabling the selection of a complementary data portfolio rather than just the single "best" seller.
>
> **Q4. Setting  for Eq. (6) coincides with the empirical distribution of the pilot dataset. Doesn't this result in privacy leakage?**
>
> We would like to clarify a misunderstanding regarding the proposed method. The equation (6) can be considered as an obfuscation mechanism as the second part is totally private $\mathbf{P}^\star(D^{\text{pi}},\gamma)\mathbf{x}^\gamma$. Specifically, the Optimal Transport plan $\mathbf{P}^\star(D^{\text{pi}},\gamma)$ has at most $n+m-1$ non-zero elements. We have provided a privacy analysis with quantifiable guarantee in the general response. The parameter $t$ explicitly controls the privacy-utility trade-off. It does not result in privacy leakage for a properly chosen $t > 0$.
>
> **Q5. Does the Eq. (9) guarantee convergence to global optimum, i.e., is Eq. (8) convex in ? Any reference is appreciated.**
>
> The convergence guarantee of our framework depends on the specific choice of the performance estimator, as we offer two variants balancing theoretical rigor and empirical precision.
>
> Specifically, if we employ the baseline estimator $f(D(N,\mathbf{p}),D^{\text{val}}) = a_1 \mathcal{W}(D(N,\mathbf{p}),D^{\text{val}}) + a_0$, maximizing the estimator is equivalent to minimizing the Wasserstein distance $\mathcal{W}$ due to the observed inverse correlation ($a_1 < 0$). According to optimal transport theory [1], the Wasserstein distance $\mathcal{W}(\sum p_i \mu_i, \nu)$ is convex with respect to the mixing weights $\mathbf{p}$. Consequently, optimizing this baseline variant constitutes a convex problem, thereby guaranteeing convergence to the global optimum.
>
> If we incorporate enhanced estimator, such as quadratic terms ($p_i^2$), it renders the objective technically non-convex—limiting theoretical guarantees to local optima. However, they are essential for capturing non-linear scaling effects and significantly improving prediction accuracy. Crucially, the optimization landscape remains well-behaved and empirically stable because it is dominated by the underlying "convex backbone" of the Wasserstein metric. This design allows our framework to provide a flexible trade-off, enabling users to prioritize either strict theoretical convergence via the baseline estimator or superior predictive performance in heterogeneous settings via the Enhanced Estimator.
>
> We have added the discussion in the Appendix of the paper.
>
> [1] Peyré, G., Cuturi, M. (2019). "Computational Optimal Transport."

---

> ### Author Response · Authors · 2025-12-03
> **Response to Reviewer HUeP [2]**
>
> **Q6.Why there exist periodic fluctuations in validation loss and accuracy in Figure 4 and 13? Which dataset do the results in Figure 4 correspond to?**
>
> We apologize for the confusion regarding the visualization. The "fluctuations" and apparent "periodicity" stem from the definition of the x-axis and the inherent stochasticity of FL training:
>
> First, the x-axis represents 64 discrete data mixing ratios , not training epochs. The apparent "periodic" pattern is a visual artifact of the lexicographical ordering of these mixing ratios (e.g., as we systematically vary the weight of Client 1 while iterating through Clients 2 and 3). As the dominance shifts cyclically between data sources with different quality, the validation performance fluctuates accordingly. Second, the fluctuations arise from the stochastic nature of FL. In highly skewed non-IID settings, FL training is inherently unstable, leading to performance variance even between similar mixing ratios.
>
> Despite these fluctuations, the key takeaway from these figures remains robust: CombineWad effectively serves as a convergence signal. As shown in the comparison, the unstable FedAvg shows no correlation, whereas the converged FedProx demonstrates a significant negative correlation between CombineWad and validation loss, validating our metric's effectiveness under stable optimization.
>
> We use CIFAR10 for both experiments.
>
> **Q7.Given the existence of a central trusted platform, is sharing perturbed/compressed pilot data (e.g., using randomized SVD) a viable alternative to the proposed method? Please provide more details of pros/cons of having a separate trusted platform.**
>
> We would like to clarify a misunderstanding regarding the trust assumption. In Section 3, "Trusted" refers to computational integrity-we assume the platform faithfully executes the calculation protocol without malicious behavior. It does not imply the platform is authorized to view raw data. Specifically, our consistent assumption throughout the paper is the standard "Honest-but-Curious" (semi-trusted) model. We will revise the terminology in the manuscript to explicitly reflect this.
>
> While SVD is indeed an efficient tool for dimensionality reduction and compression, its objective is fundamentally different from that of a privacy mechanism. The mathematical goal of SVD is to find a low-rank approximation that minimizes reconstruction error. In our setting, however, transmitting the leading singular components effectively reveals the most informative directions of variation in the data. This makes it substantially easier for an honest-but-curious platform to reconstruct sensitive attributes. Therefore, SVD is designed to preserve the essential structure of the data, whereas a privacy mechanism should deliberately obfuscate it.
>
> Our proposed mechanism directly controls the information that is revealed to the platform while still allowing it to compute the required distance and valuation quantities. For example, in the high-dimensional setting considered by [1], one projects the data onto a
> K-dimensional subspace in order to compute Wasserstein distances efficiently. Our mechanism can be applied on top of such projections to systematically obscure sensitive information in the projected domain, thereby retaining the computational benefits of subspace methods while providing stronger protection against reconstruction.
>
> [1] Subspace Robust Wasserstein Distances, ICML 2019
>
> **Q7.Please consider using principled label-noise simulation methods for the "Data Selection with Mislabeled Data" experiments in line 469, such as the pairwise flipping or symmetric flipping**
>
> We have updated the paper .

---

### Official Review · Reviewer_bt3S · 2025-11-01

**Soundness:** 3
**Presentation:** 2
**Contribution:** 3
**Rating:** 6
**Confidence:** 2

**Summary:**

The paper tackles the problem of data valuation and selection in FL-based data marketplaces, where raw data cannot be shared due to privacy constraints. The authors propose a novel framework centered on a Wasserstein-based estimator, CombineWad, that predicts model performance across unseen data combinations. The approach further introduces a privacy-preserving approximation of Wasserstein distance and leverages neural scaling laws to project performance at larger scales without full training. The framework is demonstrated on multiple datasets (CIFAR-10, MNIST, ImageNet, RSNA)

**Strengths:**

1. Introducing a federated variant of Wasserstein-based performance estimation (CombineWad) is innovative. The theoretical link between Wasserstein distance and model generalization in non-i.i.d. FL settings is interesting.
2. The experiments cover multiple FL algorithms, datasets, and heterogeneity scenarios.
3. The theoretical analysis is solid with Theorem 4.1 and Theorem 4.2.

**Weaknesses:**

1. The paper is dense and mathematically heavy. Some sections (e.g., §4.2, §C) are difficult to follow.
2. The experiments seem limited to small FL scenarios (three clients). It would strengthen the claim to test scalability (e.g., 10–20 clients with non-i.i.d. data).
3. The RSNA dataset (real-world medical dataset) is mentioned, but detailed results are missing.

**Questions:**

1. How sensitive is CombineWad to the choice of the Gaussian reference distribution?
2. Does CombineWad generalize well when the validation set is small or noisy?

---

> ### Author Response · Authors · 2025-11-30
> **Response to Reviewer bt3S**
>
> Dear Reviewer bt3S,
>
> Thanks for your constructive feedback. Hope our response could address your concerns.
>
> **W1.The paper is dense and mathematically heavy. Some sections (e.g., §4.2, §C) are difficult to follow.**
>
> Thank you for your valuable feedback. We acknowledge that certain sections of the paper may be mathematically dense and potentially challenging for some readers. However, these sections are essential for the development of the analytical tools used to analyze privacy and the approximation of Wasserstein distance. To improve clarity, we will revise these sections by providing clearer explanations of the key concepts, and we will also include a summary of the main results and their significance at the beginning of each relevant section. This should help readers better understand the technical part while maintaining the rigor of the analysis.
>
> **W2.The experiments seem limited to small FL scenarios (three clients). It would strengthen the claim to test scalability (e.g., 10–20 clients with non-i.i.d. data).**
>
> Please refer to the general response.
>
> **W3. The RSNA dataset (real-world medical dataset) is mentioned, but detailed results are missing.**
>
> The details are shown in Appendix C.3 in our original paper.
>
> **Q1. How sensitive is CombineWad to the choice of the Gaussian reference distribution?**
>
> The choice of a Gaussian reference measure is primarily for analytical tractability in deriving the closed-form privacy bound and approximation bound, but the mechanism remains effective for any distribution with finite covariance.
>
> Let $P$ be the private dataset. We construct the obfuscated proxy $\mathbf{Z}$ by interpolating between the raw data and a barycentric projection onto a stochastic reference measure $\gamma$. Specifically,$$\mathbf{z}_i = (1-t)\mathbf{x}_i + t\mathbf{Y}_i(P, \gamma),$$
> where $\mathbf{Y}_i(P, \gamma)=\big(m\,\mathbf{P}^\star(P,\gamma)\,\mathbf{x}^\gamma\big)_i$ is the barycentric map derived from the optimal transport plan between $P$ and the random $\gamma$. The randomness of $\gamma$ acts as the noise source.
>
> The core privacy mechanism relies on the variance injection from the stochastic reference measure $\gamma$. As shown in our analysis, the conditional variance of the obfuscated data $\mathbf{Z}$ (which determines the reconstruction difficulty) scales with the variance of $\gamma$. The approximation$$\operatorname{Tr}(\Sigma_{\mathbf{Z}}(P)) \approx \frac{t^2 m \operatorname{Tr}(\Sigma_\gamma)}{k_{\text{eff}}(\varepsilon)}$$holds for any reference measure $\gamma$ with a finite covariance matrix $\Sigma_\gamma$, not just Gaussian. This confirms that the privacy protection is driven by the magnitude of the injected noise ($\Sigma_\gamma$), rather than its specific shape. We provide the proof in the Appendix F.
>
> We assume a Gaussian $\gamma$ for two specific reasons:
>
> (1) Central Limit Theorem: Since the barycentric map $\mathbf{Y}_i(P,\gamma)$
> is a weighted aggregation of many independent reference points, the resulting distribution of $\mathbf{Z}$ naturally tends toward a Gaussian distribution (via CLT) even if $\gamma$ is non-Gaussian (e.g., Uniform). Thus, the Gaussian channel approximation $p(\mathbf{Z}|P)$ is physically justified.
>
> (2) Maximum Entropy: For a fixed variance, the Gaussian distribution maximizes differential entropy. Assuming $\gamma$ is Gaussian provides a standard, analytically rigorous way to quantify the information flow, often serving as a robust proxy for the "worst-case" reconstruction difficulty.
>
> **Q2. Does CombineWad generalize well when the validation set is small or noisy?**
>
> We appreciate the concern and agree that the size and quality of the validation set are critical in practice.
>
> - Small validation set: CombineWad relies on distribution-level alignment rather than fitting a high-capacity model on the validation set. The Wasserstein distance is computed between empirical measures and further smoothed via entropic regularization, so it primarily captures coarse geometric structure of the target distribution. This makes the estimator less sensitive to the exact cardinality of the validation set, as long as it provides a reasonable coverage of the target domain. Conceptually, our use of a distance signal rather than directly training on the validation se) mitigates overfitting to small validation sets, and we will clarify this limitation–robustness trade-off in the revised version.
>
> - Noisy or weakly supervised validation set: The core signal used by CombineWad can be $\textbf{label-free}$. Our experiments on Unlabeled Data Selection (RSNA, a real-world medical dataset without labels) show that CombineWad can still identify useful sources even in the complete absence of label supervision, indicating robustness to noisy or missing validation labels. We will make this connection more explicit in the paper.

---

### Official Review · Reviewer_vCLw · 2025-11-02

**Soundness:** 3
**Presentation:** 3
**Contribution:** 3
**Rating:** 4
**Confidence:** 4

**Summary:**

This paper tackles a long-standing yet unsolved challenge in federated data and model marketplaces — how to value and select data from multiple heterogeneous sources without sharing raw data.
The authors propose CombineWad, a Wasserstein-distance–based estimator that predicts model performance across unseen data combinations in a federated learning (FL) setup.

They further introduce a privacy-preserving distributed approximation of the Wasserstein distance (via interpolating measures between each party’s data and a shared Gaussian reference) and couple it with a neural scaling law–based extrapolation, enabling reliable performance prediction on larger datasets without full-scale training.

Extensive experiments across CIFAR-10, MNIST, ImageNet, and RSNA bone age datasets demonstrate that CombineWad consistently correlates with model validation performance under non-i.i.d. conditions, supports algorithm selection (e.g., FedProx vs. FedAvg), and facilitates robust data selection in label-skewed, mislabeled, and even unlabeled cases.

**Strengths:**

1. Conceptual originality: The paper combines optimal transport, privacy-preserving computation, and neural scaling laws in a new way tailored for federated marketplaces.

2. Theoretical justification: The relationship between validation loss and combined Wasserstein distance is formalized and empirically validated.

3. Strong experimental evidence:
CombineWad shows a stable negative correlation with validation performance across FL algorithms (FedAvg, FedProx, Scaffold, FedNova).
Outperforms AggWad and baseline estimators (Linear, Quadratic, Rational).
Works for noisy, label-skewed, and unlabeled data selection.

4. Practical implications: Provides a lightweight, privacy-aware tool for performance estimation and data acquisition without full retraining.

**Weaknesses:**

1. Dependence on convergence and algorithm stability:
CombineWad’s correlation with model accuracy weakens under unstable or early-stage FL training (especially for FedAvg). The method implicitly assumes good convergence, which may not always hold.

2. Scalability and sensitivity of private Wasserstein approximation:
The interpolation-based approach introduces hyperparameters whose influence on accuracy and privacy is not fully analyzed.
Its computational and communication cost under high-dimensional embeddings or large m (many clients) remains unclear.

3. Limited robustness and extrapolation analysis:
The scaling-law extrapolation achieves r^2 = 0.88 but lacks an analysis of failure cases (e.g., heavy distribution shift or long extrapolation range).

4. Lack of market/pricing integration:
The paper focuses on valuation and selection but does not connect these to pricing or trading mechanisms, which are critical for a complete “marketplace” framework.

5. Baselines and privacy comparisons:
No direct comparison with DP-based or gradient-based privacy-preserving data valuation methods (e.g., DP-OT, CLUES), which would strengthen the empirical positioning.

**Questions:**

Q1. How can CombineWad be reliably used before full convergence? Could the authors quantify its reliability or propose a convergence-detection heuristic?

Q2. How should the hyperparameters t, k, σ_γ of the private Wasserstein approximation be chosen? How sensitive are results to these choices?

Q3. What is the computational complexity and communication cost for multi-source (large m) scenarios?

Q4. How far can the scaling-law extrapolation be trusted under strong distribution shifts or limited pilot data?

Q5. Could CombineWad’s gradient signal be used to define shadow prices for data in a federated marketplace?

Q6. Have the authors evaluated privacy leakage risks (e.g., membership inference) from the exchanged interpolating measures?

**Details Of Ethics Concerns:**

Although no raw data are shared, the exchange of interpolated measures and distance matrices could expose statistical or membership information in adversarial settings.
The authors should clarify whether DP noise or quantization is applied to these shared quantities and consider including a basic privacy-leakage assessment.

---

> ### Author Response · Authors · 2025-11-30
> **Response to Reviewer vCLw [1]**
>
> Dear Reviewer vCLw,
>
> Thanks for your constructive feedback. Hope our response could address your concerns.
>
> **W1. Dependence on convergence and algorithm stability: CombineWad’s correlation with model accuracy weakens under unstable or early-stage FL training (especially for FedAvg). The method implicitly assumes good convergence, which may not always hold.**
>
> We appreciate this critical question. We acknowledge that CombineWad’s efficacy is linked to training stability, but we view this as a diagnostic strength rather than a limitation. Regarding the concern about "unrealistic" convergence assumptions, we argue that **data valuation is only meaningful when the learning process is functional**; if a model diverges (e.g., FedAvg in our experiments), the breakdown of the CombineWad-Accuracy correlation serves as a critical "Warning Signal" that the bottleneck is algorithmic instability, not data quality. To quantify reliability, we propose a practical Correlation Sanity Check. Specifically, before full-scale selection, the buyer can randomly sample $k$ small data mixtures for early-stage FL training. A strong negative correlation between CombineWad and validation accuracy confirms reliability, while a weak correlation indicates algorithmic instability, suggesting the buyer should prioritize optimizing the FL process (e.g., switching to FedProx) before data procurement.
>
> **W2. Scalability and sensitivity of private Wasserstein approximation: The interpolation-based approach introduces hyperparameters whose influence on accuracy and privacy is not fully analyzed. Its computational and communication cost under high-dimensional embeddings or large m (many clients) remains unclear.**
>
> - Hyperparameter Sensitivity: The influence of the key hyperparameters (e.g., interpolation strength $t$, reference variance $\sigma_\gamma^2$, and the number of reference points $k$) on both privacy and accuracy is formally analyzed in the general reponse and the original paper. In particular, Theorem 4.2 quantifies how these parameters affect the accuracy of the private Wasserstein approximation. We also empirically validate the theoretical findings by conducting experiments, which are presented in the Appendix. These experiments show how the performance of the mechanism scales with the hyperparameters and highlight the trade-offs between privacy and utility in practice.
>
> - Scalability and Computational Cost:  Regarding scalability, the approach has a favorable computational complexity compared to direct Wasserstein distance calculations. The primary computational overhead comes from computing the optimal transport plan $\mathbf{P}^\star(P,\gamma)$ and barycentric map $\mathbf{Y}(P,\gamma)$, which can be efficiently solved using modern solvers (e.g., Sinkhorn’s algorithm) in practice, especially when using entropic regularization.
>
> **W3. Limited robustness and extrapolation analysis: The scaling-law extrapolation achieves $r^2$ = 0.88 but lacks an analysis of failure cases (e.g., heavy distribution shift or long extrapolation range).**
>
> We provide more analysis in our updated paper draft.
>
> **W4. Lack of market/pricing integration: The paper focuses on valuation and selection but does not connect these to pricing or trading mechanisms, which are critical for a complete “marketplace” framework.**
>
> We acknowledge that a complete marketplace framework requires pricing mechanisms. However, we position CombineWad as the foundational valuation engine that supports such mechanisms.
>
> - Valuation precedes pricing: Economic pricing mechanisms (e.g., auctions, profit sharing) fundamentally rely on an accurate measure of utility. Our work solves this technical prerequisite: reliably predicting the performance contribution of data combinations without raw data access.
>
> - Flexible modular Integration: By decoupling valuation from pricing, our framework is flexible. Once the utility (marginal gain) is estimated by CombineWad, it can be plugged into various pricing strategies to complete the marketplace loop. We will clarify this scope in the revision.
>
> **W5. Baselines and privacy comparisons: No direct comparison with DP-based or gradient-based privacy-preserving data valuation methods (e.g., DP-OT, CLUES), which would strengthen the empirical positioning.**
>
> In related work in the appendix, we have discussed the difference between our paper and all related papers the reviewer mentioned. We also conduct additional experiments, in our updated paper.
>
> **Q1.How can CombineWad be reliably used before full convergence? Could the authors quantify its reliability or propose a convergence-detection heuristic?**
>
> Please refer to the response to W1.
>
> **Q2. How should the hyperparameters $t$,$k$, $\sigma_\gamma$ of the private Wasserstein approximation be chosen? How sensitive are results to these choices?**
>
> Please refer to the general response.

---

> > ### Author Response · Authors · 2025-11-30
> > **Response to Reviewer vCLw [2]**
> >
> > **Q4.Could CombineWad’s gradient signal be used to define shadow prices for data in a federated marketplace?**
> >
> > Yes. Thanks for raising this. The gradient signal derived in CombineWad can indeed be interpreted as a shadow price (marginal value) for each data source in a federated marketplace. Our performance estimator $\hat{V}$ is negatively correlated with the Wasserstein distance $\mathcal{W}$ (i.e., lower $\mathcal{W}$ implies higher $\hat{V}$).
> > By the chain rule, the marginal value of source $i$ with mixing weight $p_i$ is
> > $$\text{Shadow Price}_i = \frac{\partial \hat{V}}{\partial p_i} =\frac{\partial \hat{V}}{\partial \mathcal{W}} \cdot \frac{\partial \mathcal{W}}{\partial p_i}$$
> >
> > If $\frac{\partial \mathcal{W}}{\partial p_i} < 0$, upweighting source $i$ reduces distribution shift, yielding $\frac{\partial \hat{V}}{\partial p_i} > 0$ and thus a positive shadow price (valuable data that should be priced higher). Conversely, $\frac{\partial \mathcal{W}}{\partial p_i} > 0$ implies harmful or noisy data with negative marginal value. Importantly, these gradients come “for free” as a by-product of the OT computation, enabling efficient, dynamically updated pricing signals for each federated data provider without additional training overhead.
> >
> > **Q5. Have the authors evaluated privacy leakage risks (e.g., membership inference) from the exchanged interpolating measures?**
> >
> > Yes, we have evaluated privacy leakage risks from the exchanged interpolating measures, with a particular focus on Membership Inference Attacks, from an information-theoretic perspective.
> >
> > As detailed in our general response, our mechanism provides a quantifiable lower bound on the reconstruction risk $R$ of the private dataset $P$ from the released proxy $(\mathbf{Z} = M(P))$. Using entropy–MMSE inequalities, this reconstruction bound translates into an upper bound on the mutual information $I(P;\mathbf{Z})$ between the raw data and the exchanged interpolating measures. Prior work on privacy risk and overfitting [1] and related information-theoretic analyses show that the advantage of MIA is constrained by such information leakage. In our construction, the variance of the random reference measure $\gamma$ (and the interpolation strength) directly controls $\operatorname{Tr}(\Sigma_{\mathbf{Z}}(P))$, thereby suppressing $I(P;\mathbf{Z})$. Intuitively, a larger reconstruction error implies that $\mathbf{Z}$ does not contain enough information to reliably decide whether a specific sample belonged to $P$, which limits attacker’s MIA success rate.
> >
> > [1] Yeom et al., Privacy Risk in Machine Learning: Analyzing the Connection to Overfitting, 2018

---

### Official Review · Reviewer_xiid · 2025-11-03

**Soundness:** 2
**Presentation:** 1
**Contribution:** 2
**Rating:** 2
**Confidence:** 4

**Summary:**

This study investigates how model buyers can evaluate the values of distributed datasets and select appropriate data subsets from data sellers who act as clients in an FL setting. The authors propose a scaling-law-style method for data valuation, by which data buyers can use small-scale data samples to predict large-scale model performance with varying sizes of data subsets. Based on this approach, data buyers can select the optimal data subset sizes for data selection to maximize the model performance.

**Strengths:**

S1. This paper is well motivated and studies a significant research problem regarding data valuation.

S2. The proposed data valuation method is grounded in a solid theoretical foundation.

S3. The idea of using scaling laws for data valuation and selection is interesting.

**Weaknesses:**

W1. The writing of this paper requires improvement. First, the introduction to Wasserstein Distance in Section 2 fails to clarify its relevance to the target research problem. Second, many choices lack justification. For instance, why do the authors consider the AggWad and CombineWad calculation methods? Why is the non-IID setting examined using only three clients? Why does the non-IID setting assume non-overlapping labels (which is unrealistic in my opinion)? Third, the proposed method has not been completely introduced and explained. For example, Formula (8), which is used for performance prediction, considers only the case of fitting at two scales. Where is the predictor fitting at more scales?

W2. The authors assume that model buyers have a budget N restricting the amount of training data to purchase, which is unrealistic. Typically, model buyers have a monetary budget for purchasing training data or models. Assuming a quantity-based budget implies that all data sellers charge the same unit price, which is uncommon in a non-IID setting.

W3. The trust model in this paper is internally inconsistent. In Section 3, the authors assume that the central platform responsible for computing the Wasserstein distance is trusted. However, in Section 4.2, they consider a privacy-preserving setting in which clients' data privacy must be protected from the central platform, implying that the central platform is semi-trusted or untrusted.

W4. Some important relevant works on data valuation in FL have not been discussed, including the following papers.

[a] A principled approach to data valuation for federated learning.

[b] Secure Shapley Value for Cross-Silo Federated Learning.

[c] Fair and efficient contribution valuation for vertical federated learning.

**Questions:**

Please address the questions shown in the weaknesses part.

---

> ### Author Response · Authors · 2025-11-30
> **Response to  Reviewer xiid [1]**
>
> Dear Reviewer xiid,
>
> Thanks for your constructive feedback. Hope our response could address your concerns.
>
> **W1.1. Why considering CombinedWad and AggWad?**
>
> - Analogy to FL: Since standard FL algorithms (e.g., FedAvg) aggregate model parameters via weighted averaging, it is intuitive to approximate the global distribution shift as the weighted sum of local distances, which cooresponds to AggWad. While AggWad represents the upper bound via triangle inequality, CombineWad is designed to estimate the actual distance of the mixture distribution. It captures how different local distributions interact (and potentially compensate for each other), which a simple weighted sum ignores. We compare them to demonstrate that precise valuation requires handling the mixture geometry, not just summing local gaps. We will add more discussions in our final paper draft.
>
> - Theoretical Link: This aligns with theoretical bounds in Multi-Source Domain Adaptation (MSDA) literature (e.g., [1-3]), where the target risk is typically bounded by the weighted sum of divergences between source and target domains.
>
> [1] Optimal Transport for Multi-source Domain Adaptation under Target Shift
>
> [2] Theoretical analysis of domain adaptation with optimal transport
>
> [3] Wasserstein barycenter for multi-source domain adaptation
>
>
> **W1.2. Why considering the non-overlapping label setting?**
>
> We deliberately chose the non-overlapping label setting because it represents the most challenging scenario in Federated Learning. As the reviewer noted, this is relevant in cross-silo settings (e.g., specialized hospitals or distinct e-commerce categories). By demonstrating the effectiveness of our method in this extreme "worst-case" scenario, we establish its robustness for milder, overlapping scenarios.
>
> We utilized a three-client setup to simulate a typical cross-silo FL environment, where the number of participants is often small, but each possesses a large volume of data. We also acknowledge the scalability concern and have included additional results in the Appendix.
>
> **W1.3.Where is the predictor fitting at more scales?**
>
> The neural scaling law typically follows a power-law relationship, which is linear in the log-log scale. Theoretically, two points are the minimum required to determine the slope and intercept of this line. While Formula (8) illustrates the calculation using minimal pilot points for cost-efficiency, our framework naturally supports fitting with more scales to improve robustness against noise.
>
> **W2.The authors assume that model buyers have a budget $N$ restricting the amount of training data to purchase, which is unrealistic. Typically, model buyers have a monetary budget for purchasing training data or models. Assuming a quantity-based budget implies that all data sellers charge the same unit price, which is uncommon in a non-IID setting.**
>
>
> We acknowledge that in a real-world marketplace, buyers operate with monetary budgets and sellers may have heterogeneous pricing. However, we employ the quantity budget $N$ as a standard abstraction widely used in data selection and active learning literature to benchmark performance (e.g., [4-6]). Our work focuses on technical valuation of data(predicting the utility/performance contribution of data) rather than economic pricing mechanisms. The proposed framework solves the fundamental problem of "how to measure the value of a specific data combination," which is a prerequisite for establishing any pricing model. Once the utility is accurately estimated by our method, it can be easily integrated into variable pricing schemes (e.g., assigning higher unit prices to data with higher estimated utility). We have clarified this scope in the revised manuscript.
>
> [4] Data acquisition via experimental design for decentralized data markets. NeurIPS 2024
>
> [5] Coresets for Data-efficient Training of Machine Learning Models. ICML 2024
>
> [6] Performance Scaling via Optimal Transport: Enabling Data Selection from Partially Revealed Sources.  NeurIPS 2023

---

> ### Author Response · Authors · 2025-11-30
> **Response to Reviewer xiid [2]**
>
> **W3.The trust model in this paper is internally inconsistent. In Section 3, the authors assume that the central platform responsible for computing the Wasserstein distance is trusted. However, in Section 4.2, they consider a privacy-preserving setting in which clients' data privacy must be protected from the central platform, implying that the central platform is semi-trusted or untrusted**
>
> We would like to clarify a misunderstanding regarding the trust assumption. In Section 3, "Trusted" refers to computational integrity-we assume the platform faithfully executes the calculation protocol without malicious behavior. It does not imply the platform is authorized to view raw data. Specifically, our consistent assumption throughout the paper is the standard "Honest-but-Curious" (semi-trusted) model. We will revise the terminology in the manuscript to explicitly reflect this.
>
> **W4. Some important relevant works on data valuation in FL have not been discussed, including the following papers.
> [a] A principled approach to data valuation for federated learning. [b] Secure Shapley Value for Cross-Silo Federated Learning.[c] Fair and efficient contribution valuation for vertical federated learning.**
>
> We thank the reviewer for pointing out these important works. We will include a detailed discussion of them in our revised related work section. We distinguish our work from these approaches in the following key aspects:
>
> 1. Efficiency and Predictive Capability: While theoretically rigorous, Shapley-based methods (like [a] and [b]) often incur exponential computational costs and require full model training. Moreover, they are typically post-hoc (evaluating contribution after training) and cannot directly solve our target problem: predicting the performance of unseen data combinations to determine optimal mixing ratios before purchasing.
>
> 2. Comparison: Note that [7] have compared Wasserstein distance based method against GTG-Shapley, which represents a SOTA gradient-based Shapley approximation, demonstrating the advantage of incorporating Wasserstein distance as the metriuc over this class of methods.
>
> 3. Problem Setting: Regarding paper [c], it specifically addresses vertical FL (feature-partitioned), whereas our work focuses on Horizontal/Cross-Silo FL (sample-partitioned), making it a distinct problem setting.
>
> [7] Data valuation and detections in federated learning, CVPR 2024.

---

### Official Review · Reviewer_hxHS · 2025-11-07

**Soundness:** 3
**Presentation:** 2
**Contribution:** 2
**Rating:** 4
**Confidence:** 3

**Summary:**

This paper introduces \textbf{CombineWad}, a Wasserstein-distance–based framework for data valuation and selection in federated model marketplaces. It predicts model performance and selects optimal data combinations without accessing raw data, using a privacy-preserving Wasserstein estimator and neural scaling laws for extrapolation. Experiments on multiple datasets show strong correlation between predicted and actual performance.

**Strengths:**

1.	Tackles a real and important problem of data valuation under privacy constraints in federated learning.
2.	Provides sound theoretical grounding linking Wasserstein distance to validation performance.
3.	Empirical results show promising predictive accuracy and practical relevance

**Weaknesses:**

1.	Some important references are missed in the article, e.g. “we provide Table ??, which summarizes the important notations used throughout this work.” In the third Section.
2.	Experiments are mostly synthetic or small-scale, which lack the robustness needed to demonstrate real-world scalability.
3.	Computational and communication overheads of the Wasserstein estimator in high-dimensional data or with many clients are not analyzed.

**Questions:**

1. How does the proposed private Wasserstein estimator guarantee privacy formally (e.g., under $\epsilon$-DP or other quantifiable bounds)?
2. Could the estimator generalize to vertical FL or cross-domain settings?
3. How sensitive is the method to pilot data size? Would a small pilot set (e.g., 1% of data) still yield accurate predictions?

---

> ### Author Response · Authors · 2025-11-30
> **Response to Reviewer hxHS**
>
> Dear Reviewer hxHS,
>
> Thanks for your constructive feedback. Hope our response could address your concerns.
>
> **W1. Experiments are mostly synthetic or small-scale, which lack the robustness needed to demonstrate real-world scalability.**
>
> We conducted an experiment on the RSNA dataset (real-world medical dataset), and details are shown in Appendix C.3 in our original paper. For additional experiments, please refer to the general response.
>
> **W2. Computational and communication overheads of the Wasserstein estimator in high-dimensional data or with many clients are not analyzed.**
>
> In terms of computational complexity, we need to compute  $2N+1$  OT plans in total, based on the network simplex is $\mathcal{O}((n+m)nm\log(n+m))$
>
> **Q1.How does the proposed private Wasserstein estimator guarantee privacy formally (e.g., under -DP or other quantifiable bounds)?**
>
> Please refer to the general response.
>
> **Q2.Could the estimator generalize to vertical FL or cross-domain settings?**
>
> Currently our method only support to the horizontal FL, as Wasserstein distance is a metric between two data distributions, and it is challenging to measure the "distance" between non-overlapping features.
>
> **Q3.How sensitive is the method to pilot data size? Would a small pilot set (e.g., 1$\%$ of data) still yield accurate predictions?**
>
> Our empirical results show that the method remains effective even with small pilot sets. Please refer to additional experiments in the updated paper draft.
>
> As a global distributional metric, Wasserstein distance captures the geometry of the data distribution effectively even with limited samples. (2) Various experiments confirm that the data mixing ratios selected based on small pilot sets consistently yield high performance when scaled up. This indicates that while the pilot size is small, it is sufficient to identify the "good" data combinations.
>
> We acknowledge that while the relative ranking of data sources is robust (enabling effective selection), the absolute predicted accuracy via the estimator may exhibit some variance due to finite-sample approximations. We believe exploring more sophisticated mapping functions to further reduce this variance is a valuable direction for future work.

---

### Author Response · Authors · 2025-11-30
**General Response**

Dear Reviewers and ACs,

Thanks for your constructive feedback. Hope our response could address your concerns.

**General Response 1. Privacy Analysis with Quantifiable Guarantee**

Our proposed method is not designed to satisfy worst-case $(\varepsilon,\delta)$-Differential Privacy (DP). In our setting, enforcing worst-case DP fundamentally conflicts with the goal of obtaining a high-utility, geometry-preserving approximation of Wasserstein distances. In particular, the amount of noise required by DP can introduce substantial distortion in the distance estimates, as empirically observed in our updated paper draft. This level of distortion is incompatible with our target applications, where preserving the underlying Wasserstein geometry is a primary objective.

Instead, we adopt a different and more tailored privacy notion, which we make formal below: a \emph{reconstruction privacy} (Bayesian risk) guarantee for the raw data $P$. Our mechanism leverages the geometric structure of Wasserstein geodesics while injecting randomness through a latent reference measure. Concretely, even if the reference measure $\gamma$ is public, an adversary attempting to recover $P$ from the proxy $\eta_P$ must solve a highly non-linear and computationally demanding \emph{inverse optimal transport} problem, because the optimal transport plan $\mathbf{P}^\star(P,\gamma)$ is never revealed.

Moreover, the core randomization in our construction is a one-time, shared sampling of the reference measure $\gamma$. Since $\gamma$ is stochastic, the proxy $\eta_P$ is itself a random variable. This induces a form of plausible deniability: observing a particular realization of $\eta_P$ does not allow an adversary to reliably distinguish whether it originated from $P_1$ or $P_2$, as both inputs can produce compatible outputs under different realizations of the random reference measure.

We formalize this intuition and provide a formal privacy analysis based on Reconstruction Risk ($R$) of our geometric obfuscation mechanism. \textbf{We respectfully refer the reviewer to Appendix F for the complete proofs and derivations for the following conclusion.}

Let $P$ be the private dataset. We construct the obfuscated proxy $\mathbf{Z}$ by interpolating between the raw data and a barycentric projection onto a stochastic reference measure $\gamma \sim \mathcal{N}(\mu, \sigma_\gamma^2 I)$. Specifically,$$\mathbf{z}_i = (1-t)\mathbf{x}_i + t\mathbf{Y}_i(P, \gamma),$$
where $\mathbf{Y}_i(P, \gamma)=\big(m\,\mathbf{P}^\star(P,\gamma)\,\mathbf{x}^\gamma\big)_i$ is the barycentric map derived from the optimal transport plan between $P$ and the random $\gamma$. The randomness of $\gamma$ acts as the noise source.


We derive a rigorous lower bound on the reconstruction risk $R$ (defined as the MMSE of recovering $P$ from $\mathbf{Z}$). We prove that:$$R \gtrsim f\left(\frac{t^2 \cdot m \cdot d \cdot \sigma_\gamma^2}{k_{\text{eff}}(\varepsilon)}\right)$$This bound establishes a formal information-theoretic barrier against reconstruction, and it provides a clear mechanism to control the privacy-utility trade-off:

-  Push-forward parameter ($t$): Privacy improves quadratically with $t$ ($R \propto t^2$).

- Reference Noise ($\sigma_\gamma^2$): Privacy improves linearly with the variance of the random reference measure.

- Effective Support ($k_{\text{eff}}$): Sparsity in the transport plan (controlled by regularization $\varepsilon$) further enhances reconstruction difficulty.

It is worthy to note that this framework formally quantifies the trade-offs between utility (Theorem 1) and reconstruction privacy ($R$), which also aligns with the result in our original paper. Our proof is updated in Appendix F.

---

### Meta-Review · Area_Chair_HZsm · 2026-01-07

**Summary:**

While the paper is investigating an important topic and proposing an interesting approach, several of the reviewers concerns are not fully addressed (in particular regarding scalability but also regarding baselines; for more see below). The updates to the paper would also require an additional review. Considering how reviewers would view the paper after the rebuttal, I would assume the paper to be still borderline.

**Reviewer Concerns:**

Reviewer hxHS
- Experiments are mostly synthetic/small-scale and lack of demonstration of real-world scalability
  -  Partially addressed by experiments on RSNA; the authors also refer to the general response in their rebuttal but there is no additional information provided there
- Computational/communication overheads of private Wasserstein in high dimensions/many clients not analyzed
  - Adressed
- Formal privacy guarantees
  - Addressed via reconstruction-privacy analysis present in Appendix F)
- Generalization to vertical FL/cross-domain
  - Addressed but limited to horizontal FL as acknowledged by the authors
- Sensitivity to pilot size
  - Partially addressed via empirical results and added experiments; details not clear from rebuttal.

Reviewer xiid
- Writing clarity, in particular relevance of WD, justification of design choices, and multi-scale predictor
  - Partially addressed via explanations and promise to add more discussions to the paper
- Quantity budget vs. monetary budget realism
  - Partially addressed by justification as standard abstraction and scope clarification; unclear whether this would would satisfy the reviewer
- Trust model inconsistency
  - Addressed, misconception of the reviewer
- Missing related FL data valuation works
  - Partially addressed by promising to include a discussion of some of the proposed works

Reviewer vCLw
- Dependence on convergence/stability
  - Partially addressed (proposed a "Correlation Sanity Check" heuristic)
- Hyperparameter sensitivity and scalability costs
  - Partially addressed by references to analysis and modern solvers; unclear whether this would have sufficient depths to convince the reviewer
- Robustness/failure cases of scaling-law extrapolation
  - Partially addressed by analysis in updated paper draft
- Lack of market/pricing integration
  - Partially addressed by clarifying the scope of the paper
- Baselines/privacy comparisons (DP-OT, CLUES)
  - Addressed by discussing the differences to the submitted paper and additional experiements
- Privacy-leakage evaluation
  - Addressed by theoretical analysis

Reviewer bt3S
- Limited to small-scale scenarios
  - Partially addressed by general response
- Detailed results for RSNA dataset are missing
  - Partially addressed, the authors referred to appendix C.3 but it is unclear whether this would satisfy the reviewer.
- Sensitivity to Gaussian reference
  - Addressed.
- Generalization with small/noisy validation
  - Addressed

Reviewer HUeP
- Strict central platform assumption
  - Addressed by clarifying the honest-but-curious assumption
- Empirical validation insufficient
  - Not clearly addressed, still standing (overlap with questions from other reviewers)
- Need diverse/adaptive aggregation baselines (TERM, AAggFF)
  - Not clearly addressed, still standing
- Existence/size of validation data; parameter sizing in Theorem 4.2 and communication cost
  - Not clearly addressed
- Other issues
  - Addressed

**Reviewer Scores:**

* Reviewer hxHS might change the score to borderline accept
* Reviewer xiid might increase the score slightly but as not all concerns were fully addressed would likely still lean towards rejection
* Reviewer vCLw might change the score to borderline accept
* Reviewer bt3S would likely not change their assessment as they were already positive about the paper and the deteails regarding RSNA might still be considered insufficient
* Reviewer HUeP would likely not change their score as the several concerns are not clearly addressed

---

### Decision · Program_Chairs · 2026-01-26

Reject